# Understanding and Robustifying Differentiable Architecture Search

**Arber Zela**[1]**, Thomas Elsken**[2,1]**, Tonmoy Saikia**[1]**, Yassine Marrakchi**[1]**,
Thomas Brox**[1] **& Frank Hutter**[1,2]
[1]Department of Computer Science, University of Freiburg
`{zelaa, saikiat, marrakch, brox, fh}@cs.uni-freiburg.de`
[2]Bosch Center for Artificial Intelligence
`Thomas.Elsken@de.bosch.com`

## Abstract

Differentiable Architecture Search (DARTS) has attracted a lot of attention due to its simplicity and small search costs achieved by a continuous relaxation and an approximation of the resulting bi-level optimization problem. However, DARTS does not work robustly for new problems: we identify a wide range of search spaces for which DARTS yields degenerate architectures with very poor test performance. We study this failure mode and show that, while DARTS successfully minimizes validation loss, the found solutions generalize poorly when they coincide with high validation loss curvature in the architecture space. We show that by adding one of various types of regularization we can robustify DARTS to find solutions with less curvature and better generalization properties. Based on these observations, we propose several simple variations of DARTS that perform substantially more robustly in practice. Our observations are robust across five search spaces on three image classification tasks and also hold for the very different domains of disparity estimation (a dense regression task) and language modelling.

## 1 Introduction

Neural Architecture Search (NAS), the process of automatically designing neural network architectures, has recently attracted attention by achieving state-of-the-art performance on a variety of tasks (Zoph & Le, 2017; Real et al., 2019). *Differentiable architecture search* (DARTS) (Liu et al., 2019) significantly improved the efficiency of NAS over prior work, reducing its costs to the same order of magnitude as training a single neural network. This expanded the scope of NAS substantially, allowing it to also be applied on more expensive problems, such as semantic segmentation (Chenxi et al., 2019) or disparity estimation (Saikia et al., 2019).

However, several researchers have also reported DARTS to *not* work well, in some cases even no better than random search (Li & Talwalkar, 2019; Sciuto et al., 2019). Why is this? How can these seemingly contradicting results be explained? The overall goal of this paper is to understand and overcome such failure modes of DARTS. To this end, we make the following contributions:

1. We identify 12 NAS benchmarks based on four search spaces in which standard DARTS yields degenerate architectures with poor test performance across several datasets (Section 3).
2. By computing the eigenspectrum of the Hessian of the validation loss with respect to the architectural parameters, we show that there is a strong correlation between its dominant eigenvalue and the architecture's generalization error. Based on this finding, we propose a simple variation of DARTS with early stopping that performs substantially more robustly (Section 4).
3. We show that, related to previous work on sharp/flat local minima, regularizing the inner objective of DARTS more strongly allows it to find solutions with smaller Hessian spectrum and better generalization properties. Based on these insights, we propose two practical robustifications of DARTS that overcome its failure modes in all our 12 NAS benchmarks (Section 5).

Our findings are robust across a wide range of NAS benchmarks based on image recognition and also hold for the very different domains of language modelling (PTB) and disparity estimation. They

consolidate the findings of the various results in the literature and lead to a substantially more robust version of DARTS. We provide our implementation and scripts to facilitate reproducibility[1].

# 2 BACKGROUND AND RELATED WORK

## 2.1 RELATION BETWEEN FLAT/SHARP MINIMA AND GENERALIZATION PERFORMANCE

Already Hochreiter & Schmidhuber (1997) observed that flat minima of the training loss yield better generalization performance than sharp minima. Recent work (Keskar et al., 2016; Yao et al., 2018) focuses more on the settings of large/small batch size training, where observations show that small batch training tends to get attracted to flatter minima and generalizes better. Similarly, Nguyen et al. (2018) observed that this phenomenon manifests also in the hyperparameter space. They showed that whenever the hyperparameters overfit the validation data, the minima lie in a sharper region of the space. This motivated us to conduct a similar analysis in the context of differentiable architecture search later in Section 4.1, where we see the same effect in the space of neural network architectures.

## 2.2 BI-LEVEL OPTIMIZATION

We start by a short introduction of the bi-level optimization problem (Colson et al., 2007). These are problems which contain two optimization tasks, nested within each other.

**Definition 2.1.** Given the outer objective function $F : \mathbb{R}^P \times \mathbb{R}^N \to \mathbb{R}$ and the inner objective function $f : \mathbb{R}^P \times \mathbb{R}^N \to \mathbb{R}$, the bi-level optimization problem is given by

$$\min_{y \in \mathbb{R}^P} F(y, \theta^*(y)) \tag{1}$$

$$s.t. \quad \theta^*(y) \in \arg\min_{\theta \in \mathbb{R}^N} f(y, \theta), \tag{2}$$

where $y \in \mathbb{R}^P$ and $\theta \in \mathbb{R}^N$ are the outer and inner variables, respectively. One may also see the bi-level problem as a constrained optimization problem, with the inner problem as a constraint.

In general, even in the case when the inner objective (2) is strongly convex and has an unique minimizer $\theta^*(y) = \arg\min_{\theta \in \mathbb{R}^N} f(y, \theta)$, it is not possible to directly optimize the outer objective (1). A possible method around this issue is to use the implicit function theorem to retrieve the derivative of the solution map (or response map) $\theta^*(y) \in \mathbb{F} \subseteq \mathbb{R}^N$ w.r.t. $y$ (Bengio, 2000; Pedregosa, 2016; Beirami et al., 2017). Another strategy is to approximate the inner problem with a dynamical system (Domke, 2012; Maclaurin et al., 2015; Franceschi et al., 2017; 2018), where the optimization dynamics could, e.g., describe gradient descent. In the case that the minimizer of the inner problem is unique, under some conditions the set of minimizers of this approximate problem will indeed converge to the minimizers of the bilevel problem (1) (see Franceschi et al. (2018)).

## 2.3 NEURAL ARCHITECTURE SEARCH

Neural Architecture Search (NAS) denotes the process of automatically designing neural network architectures in order to overcome the cumbersome trial-and-error process when designing architectures manually. We briefly review NAS here and refer to the recent survey by Elsken et al. (2019b) for a more thorough overview. Prior work mostly employs either reinforcement learning techniques (Baker et al., 2017a; Zoph & Le, 2017; Zhong et al., 2018; Zoph et al., 2018) or evolutionary algorithms (Stanley & Miikkulainen, 2002; Liu et al., 2018b; Miikkulainen et al., 2017; Real et al., 2017; 2019) to optimize the discrete architecture space. As these methods are often very expensive, various works focus on reducing the search costs by, e.g., employing network morphisms (Cai et al., 2018a;b; Elsken et al., 2017; 2019a), weight sharing within search models (Saxena & Verbeek, 2016; Bender et al., 2018; Pham et al., 2018) or multi-fidelity optimization (Baker et al., 2017b; Falkner et al., 2018; Li et al., 2017; Zela et al., 2018), but their applicability still often remains restricted to rather simple tasks and small datasets.

---

[1] https://github.com/automl/RobustDARTS

## 2.4 DIFFERENTIABLE ARCHITECTURE SEARCH (DARTS)

A recent line of work focuses on relaxing the discrete neural architecture search problem to a continuous one that can be solved by gradient descent (Liu et al., 2019; Xie et al., 2019; Casale et al., 2019; Cai et al., 2019). In DARTS (Liu et al., 2019), this is achieved by simply using a weighted sum of possible candidate operations for each layer, whereas the real-valued weights then effectively parametrize the network's architecture. We will now review DARTS in more detail, as our work builds directly upon it.

**Continuous relaxation of the search space.** In agreement with prior work (Zoph et al., 2018; Real et al., 2019), DARTS optimizes only substructures called cells that are stacked to define the full network architecture. Each cell contains $N$ nodes organized in a directed acyclic graph. The graph contains two inputs nodes (given by the outputs of the previous two cells), a set of intermediate nodes, and one output node (given by concatenating all intermediate nodes). Each intermediate node $x^{(j)}$ represents a feature map. See Figure 1 for an illustration of such a cell. Instead of applying a single operation to a specific node during architecture search, Liu et al. (2019) relax the decision which operation to choose by computing the intermediate node as a mixture of candidate operations, applied to predecessor nodes $x^{(i)}, i < j$, $x^{(j)} = \sum_{i<j} \sum_{o \in \mathcal{O}} \frac{\exp(\alpha_o^{i,j})}{\sum_{o' \in \mathcal{O}} \exp(\alpha_{o'}^{i,j})} o\left(x^{(i)}\right)$, where $\mathcal{O}$ denotes the set of all candidate operations (e.g., $3 \times 3$ convolution, skip connection, $3 \times 3$ max pooling, etc.) and $\alpha = (\alpha_o^{i,j})_{i,j,o}$ serves as a real-valued parameterization of the architecture.

**Gradient-based optimization of the search space.** DARTS then optimizes both the weights of the search network (often called the *weight-sharing* or *one-shot* model, since the weights of all individual subgraphs/architectures are shared) and architectural parameters by alternating gradient descent. The network weights and the architecture parameters are optimized on the training and validation set, respectively. This can be interpreted as solving the bi-level optimization problem (1), (2), where $F$ and $f$ are the validation and training loss, $\mathcal{L}_{valid}$ and $\mathcal{L}_{train}$, respectively, while $y$ and $\theta$ denote the architectural parameters $\alpha$ and network weights $w$, respectively. Note that DARTS only approximates the lower-level solution by a single gradient step (see Appendix A for more details).

At the end of the search phase, a discrete cell is obtained by choosing the $k$ most important incoming operations for each intermediate node while all others are pruned. Importance is measured by the operation weighting factor $\frac{\exp(\alpha_o^{i,j})}{\sum_{o' \in \mathcal{O}} \exp(\alpha_{o'}^{i,j})}$.

## 3 WHEN DARTS FAILS

We now describe various search spaces and demonstrate that standard DARTS fails on them. We start with four search spaces similar to the original CIFAR-10 search space but simpler, and evaluate across three different datasets (CIFAR-10, CIFAR-100 and SVHN). They are quite standard in that they use the same macro architecture as the original DARTS paper (Liu et al., 2018a), consisting of normal and reduction cells; however, they only allow a subset of operators for the cell search space:

**S1:** This search space uses a different set of only two operators per edge, which we identified using an offline process that iteratively dropped the operations from the original DARTS search space with the least importance. This pre-optimized space has the advantage of being quite small while still including many strong architectures. We refer to Appendix B for details on its construction and an illustration (Figure 9).

**S2:** In this space, the set of candidate operations per edge is $\{3 \times 3 \ SepConv, SkipConnect\}$. We choose these operations since they are the most frequent ones in the discovered cells reported by Liu et al. (2019).

**S3:** In this space, the set of candidate operations per edge is $\{3 \times 3 \ SepConv, SkipConnect, Zero\}$, where the *Zero* operation simply replaces every value in the input feature map by zeros.

**S4:** In this space, the set of candidate operations per edge is $\{3 \times 3 \ SepConv, Noise\}$, where the *Noise* operation simply replaces every value from the input feature map by noise $\epsilon \sim \mathcal{N}(0,1)$. This is the only space out of S1-S4 that is not a strict subspace of the

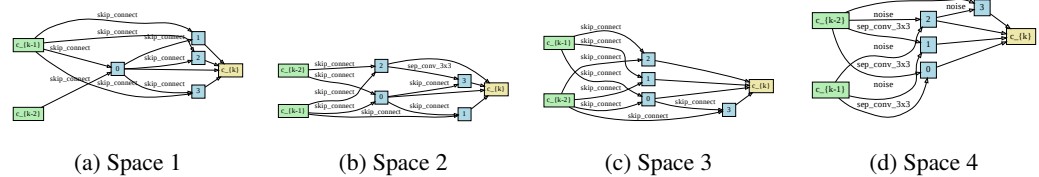

| (a) Space 1 | (b) Space 2 | (c) Space 3 | (d) Space 4 |

Figure 1: The poor cells standard DARTS finds on spaces S1-S4. For all spaces, DARTS chooses mostly parameter-less operations (skip connection) or even the harmful $Noise$ operation. Shown are the normal cells on CIFAR-10; see Appendix G for reduction cells and other datasets.

original DARTS space; we intentionally added the *Noise* operation, which actively harms performance and should therefore not be selected by DARTS.

We ran DARTS on each of these spaces, using exactly the same setup as Liu et al. (2019). Figure 1 shows the poor cells DARTS selected on these search spaces for CIFAR-10 (see Appendix G for analogous results on the other datasets). Already visually, one might suspect that the found cells are suboptimal: the parameter-less skip connections dominate in almost all the edges for spaces S1-S3, and for S4 even the harmful *Noise* operation was selected for five out of eight operations. Table 1 (first column) confirms the very poor performance standard DARTS yields on all of these search spaces and on different datasets. We note that Liu et al. (2019) and Xie et al. (2019) argue that the *Zero* operation can help to search for the architecture topology and choice of operators jointly, but in our experiments it did not help to reduce the importance weight of the skip connection (compare Figure 1b vs. Figure 1c).

We emphasize that search spaces S1-S3 are very natural, and, as strict subspaces of the original space, should merely be easier to search than that. They are in no way special or constructed in an adversarial manner. Only S4 was constructed specifically to show-case the failure mode of DARTS selecting the obviously suboptimal *Noise* operator.

**S5: Very small search space with known global optimum.** Knowing the global minimum has the advantage that one can benchmark the performance of algorithms by measuring the regret of chosen points with respect to the known global minimum. Therefore, we created another search space with only one intermediate node for both normal and reduction cells, and 3 operation choices in each edge, namely $3 \times 3$ *SepConv*, *SkipConnection*, and $3 \times 3$ *MaxPooling*. The total number of possible architectures in this space is 81, all of which we evaluated a-priori. We dub this space **S5**.

We ran DARTS on this search space three times for each dataset and compared its result to the baseline of Random Search with weight sharing (RS-ws) by Li & Talwalkar (2019). Figure 2 shows the test regret of the architectures selected by DARTS (blue) and RS-ws (green) throughout the search. DARTS manages to find an architecture close to the global minimum, but around epoch 40 the test performance deteriorated. Note that the search model validation error (dashed red line) did not deteriorate but rather converged, indicating that the architectural parameters are overfitting to the validation set. In contrast, RS-ws stays relatively constant throughout the search; when evaluating only the final architecture found, RS-ws indeed outperformed DARTS.

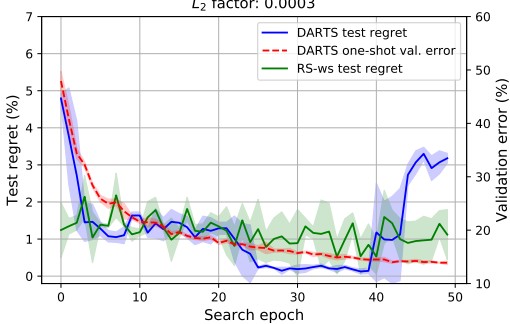

Figure 2: Test regret of found architectures and validation error of the search model when running DARTS on S5 and CIFAR-10. DARTS finds the global minimum but starts overfitting the architectural parameters to the validation set in the end.

**S6: encoder-decoder architecture for disparity estimation.** To study whether our findings generalize beyond image recognition, we also analyzed a search space for a very different problem: finding encoder-decoder architectures for the dense regression task of disparity estimation; please

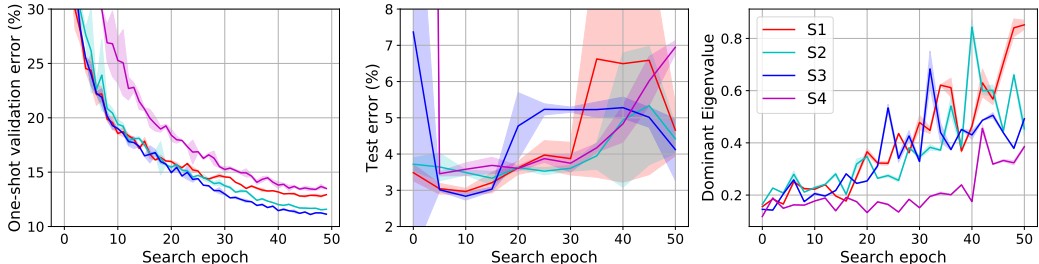

Figure 3: *(left)* validation error of search model; *(middle)* test error of the architectures deemed by DARTS optimal *(right)* dominant eigenvalue of $\nabla^2_\alpha \mathcal{L}_{valid}$ throughout DARTS search. Solid line and shaded areas show mean and standard deviation of 3 independent runs. All experiments conducted on CIFAR-10.

refer to Appendix E for details. We base this search space on AutoDispNet (Saikia et al., 2019), which used DARTS for a space containing *normal*, *downsampling* and *upsampling* cells. We again constructed a reduced space. Similarly to the image classification search spaces, we found the normal cell to be mainly composed of parameter-less operations (see Figure 25 in Appendix G). As expected, this causes a large generalization error (see first row in Table 2 of our later experiments).

# 4 THE ROLE OF DOMINANT EIGENVALUES OF $\nabla^2_\alpha \mathcal{L}_{valid}$

We now analyze *why* DARTS fails in all these cases. Motivated by Section 2.1, we will have a closer look at the largest eigenvalue $\lambda^\alpha_{max}$ of the Hessian matrix of validation loss $\nabla^2_\alpha \mathcal{L}_{valid}$ w.r.t. the architectural parameters $\alpha$.

## 4.1 LARGE ARCHITECTURAL EIGENVALUES AND GENERALIZATION PERFORMANCE

One may hypothesize that DARTS performs poorly because its approximate solution of the bi-level optimization problem by iterative optimization fails, but we actually observe validation errors to progress nicely: Figure 3 (left) shows that the search model validation error converges in all cases, even though the cell structures selected here are the ones in Figure 1.

Rather, the architectures DARTS finds do not generalize well. This can be seen in Figure 3 (middle). There, every 5 epochs, we evaluated the architecture deemed by DARTS to be optimal according to the $\alpha$ values. Note that whenever evaluating on the test set, we retrain from scratch the architecture obtained after applying the *argmax* to the architectural weights $\alpha$. As one can notice, the architectures start to degenerate after a certain number of search epochs, similarly to the results shown in Figure 2. We hypothesized that this might be related to sharp local minima as discussed in Section 2.1. To test this hypothesis, we computed the full Hessian $\nabla^2_\alpha \mathcal{L}_{valid}$ of the validation loss w.r.t. the architectural parameters on a randomly sampled mini-batch. Figure 3 (right) shows that the dominant eigenvalue $\lambda^\alpha_{max}$ (which serves as a proxy for the sharpness) indeed increases

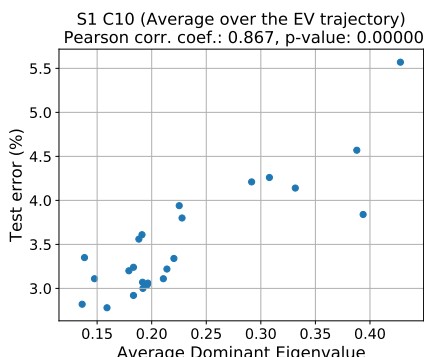

Figure 4: Correlation between dominant eigenvalue of $\nabla^2_\alpha \mathcal{L}_{valid}$ and test error of corresponding architectures.

in standard DARTS, along with the test error (middle) of the final architectures, while the validation error still decreases (left). We also studied the correlation between $\lambda^\alpha_{max}$ and test error more directly, by measuring these two quantities for 24 different architectures (obtained via standard DARTS and the regularized versions we discuss in Section 5). For the example of space S1 on CIFAR-10, Figure 4 shows that $\lambda^\alpha_{max}$ indeed strongly correlates with test error (with a Pearson correlation coefficient of 0.867).

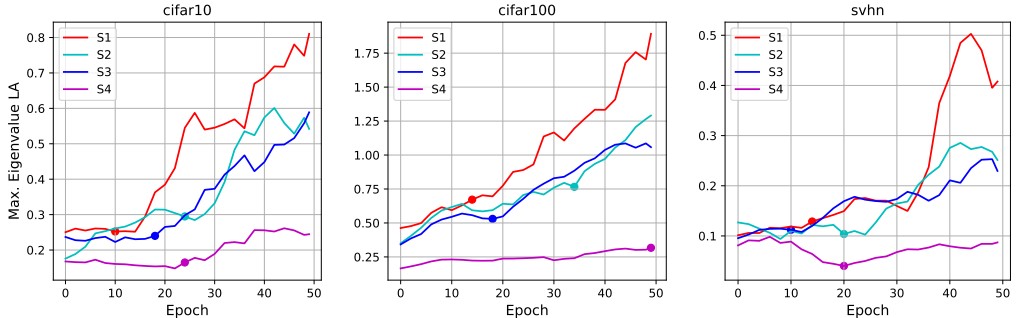

Figure 6: Local average (LA) of the dominant eigenvalue $\lambda_{max}^{\alpha}$ throughout DARTS search. Markers denote the early stopping point based on the criterion in Section 4.3. Each line also corresponds to one of the runs in Table 1.

## 4.2 LARGE ARCHITECTURAL EIGENVALUES AND PERFORMANCE DROP AFTER PRUNING

One reason why DARTS performs poorly when the architectural eigenvalues are large (and thus the minimum is sharp) might be the pruning step at the end of DARTS: the optimal, continuous $\alpha^*$ from the search is pruned to obtain a discrete $\alpha^{disc}$, somewhere in the neighbourhood of $\alpha^*$. In the case of a sharp minimum $\alpha^*$, $\alpha^{disc}$ might have a loss function value significantly higher than the minimum $\alpha^*$, while in the case of a flat minimum, $\alpha^{disc}$ is expected to have a similar loss function value. This is hypothetically illustrated in Figure 5a, where the *y-axis* indicates the search model validation loss and the *x-axis* the $\alpha$ values.

To investigate this hypothesis, we measured the performance drop: $\mathcal{L}_{valid}(\alpha^{disc}, w^*) - \mathcal{L}_{valid}(\alpha^*, w^*)$ w.r.t. to the search model weights incurred by this discretization step and correlated it with $\lambda_{max}^{\alpha}$. The results in Figure 5b show that, indeed, low curvature never led to large performance drops (here we actually compute the accuracy drop rather than the loss function difference, but we observed a similar relationship). Having identified this relationship, we now move on to avoid high curvature.

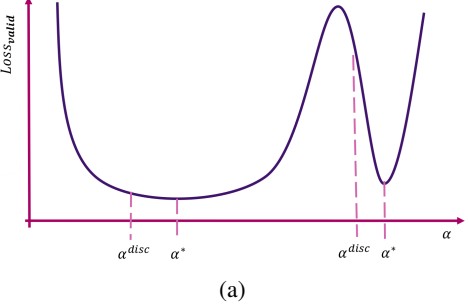

(a)

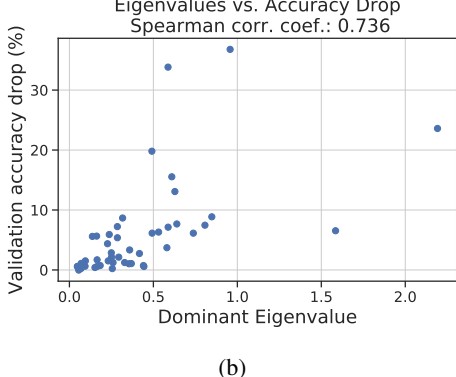

(b)

Figure 5: (a) Hypothetical illustration of the loss function change in the case of flat vs. sharp minima. (b) Drop in accuracy after discretizing the search model vs. the sharpness of minima (by means of $\lambda_{max}^{\alpha}$).

## 4.3 EARLY STOPPING
BASED ON LARGE EIGENVALUES OF $\nabla_{\alpha}^{2}\mathcal{L}_{valid}$

We propose a simple early stopping methods to avoid large curvature and thus poor generalization. We emphasize that simply stopping the search based on validation performance (as one would do in the case of training a network) does *not* apply here as NAS directly optimizes validation performance, which – as we have seen in Figure 2 – keeps on improving.

Instead, we propose to track $\lambda_{max}^{\alpha}$ over the course of architecture search and stop whenever it increases too much. To implement this idea, we use a simple heuristic that worked off-the-shelf without any tuning. Let $\overline{\lambda}_{max}^{\alpha}(i)$ denote the value of $\lambda_{max}^{\alpha}$ smoothed over $k = 5$ epochs around $i$; then, we stop if $\overline{\lambda}_{max}^{\alpha}(i-k)/\overline{\lambda}_{max}^{\alpha}(i) < 0.75$ and return the architecture from epoch $i - k$.

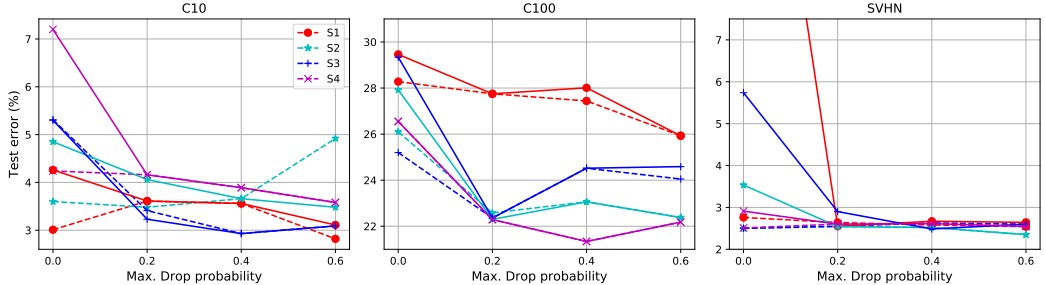

Figure 7: Effect of regularization strength via ScheduledDropPath (during the search phase) on the test performance of DARTS (solid lines) and DARTS-ES (dashed-lines). Results for each of the search spaces and datasets.

By this early stopping heuristic, we do not only avoid exploding eigenvalues, which are correlated with poor generalization (see Figure 4), but also shorten the time of the search.

Table 1 shows the results for running DARTS with this early stopping criterion (DARTS-ES) across S1-S4 and all three image classification datasets. Figure 6 shows the local average of the eigenvalue trajectory throughout the search and the point where the DARTS search early stops for each of the settings in Table 1. Note that we never use the test data when applying the early stopping mechanism. Early stopping significantly improved DARTS for all settings without ever harming it.

Table 1: Performance of DARTS and DARTS-ES. (mean $\pm$ std for 3 runs each).

| Benchmark | | DARTS | DARTS-ES |
|---|---|---|---|
| C10 | S1 | $4.66 \pm 0.71$ | $\mathbf{3.05 \pm 0.07}$ |
| | S2 | $4.42 \pm 0.40$ | $\mathbf{3.41 \pm 0.14}$ |
| | S3 | $4.12 \pm 0.85$ | $\mathbf{3.71 \pm 1.14}$ |
| | S4 | $6.95 \pm 0.18$ | $\mathbf{4.17 \pm 0.21}$ |
| C100 | S1 | $29.93 \pm 0.41$ | $\mathbf{28.90 \pm 0.81}$ |
| | S2 | $28.75 \pm 0.92$ | $\mathbf{24.68 \pm 1.43}$ |
| | S3 | $29.01 \pm 0.24$ | $\mathbf{26.99 \pm 1.79}$ |
| | S4 | $24.77 \pm 1.51$ | $\mathbf{23.90 \pm 2.01}$ |
| SVHN | S1 | $9.88 \pm 5.50$ | $\mathbf{2.80 \pm 0.09}$ |
| | S2 | $3.69 \pm 0.12$ | $\mathbf{2.68 \pm 0.18}$ |
| | S3 | $4.00 \pm 1.01$ | $\mathbf{2.78 \pm 0.29}$ |
| | S4 | $2.90 \pm 0.02$ | $\mathbf{2.55 \pm 0.15}$ |

## 5 REGULARIZATION OF INNER OBJECTIVE IMPROVES GENERALIZATION OF ARCHITECTURES

As we saw in Section 4.1, sharper minima (by means of large eigenvalues) of the validation loss lead to poor generalization performance. In our bi-level optimization setting, the outer variables' trajectory depends on the inner optimization procedure. Therefore, we hypothesized that modifying the landscape of the inner objective $\mathcal{L}_{train}$ could redirect the outer variables $\alpha$ to flatter areas of the architectural space. We study two ways of regularization (data augmentation in Section 5.1 and $L_2$ regularization in Section 5.2) and find that both, along with the early stopping criterion from Section 4.3, make DARTS more robust in practice. We emphasize that we *do not* alter the regularization of the final training and evaluation phase, but solely that of the search phase. The setting we use for all experiments in this paper to obtain the final test performance is described in Appendix C.

### 5.1 REGULARIZATION VIA DATA AUGMENTATION

We first investigate the effect of regularizing via data augmentation, namely masking out parts of the input and intermediate feature maps via Cutout (CO, DeVries & Taylor (2017)) and ScheduledDropPath (DP, Zoph et al. (2018)) (ScheduledDropPath is a regularization technique, but we list it here since we apply it together with Cutout), respectively, during architecture search. We ran DARTS with CO and DP (with and without our early stopping criterion, DARTS-ES) with different maximum DP probabilities on all three image classification datasets and search spaces S1-S4.

Figure 7 summarizes the results: regularization improves the test performance of DARTS and DARTS-ES in all cases, sometimes very substantially, and at the same time kept the dominant eigenvalue relatively low (Figure 13). This also directly results in smaller drops in accuracy after pruning, as discussed in Section 4.2; indeed, the search runs plotted in Figure 5b are the same as in this section. Figure 17 in the appendix explicitly shows how regularization relates to the accuracy drops. We also refer to further results in the appendix: Figure 11 (showing test vs. validation error) and Table 5 (showing that overfitting of the architectural parameters is reduced).

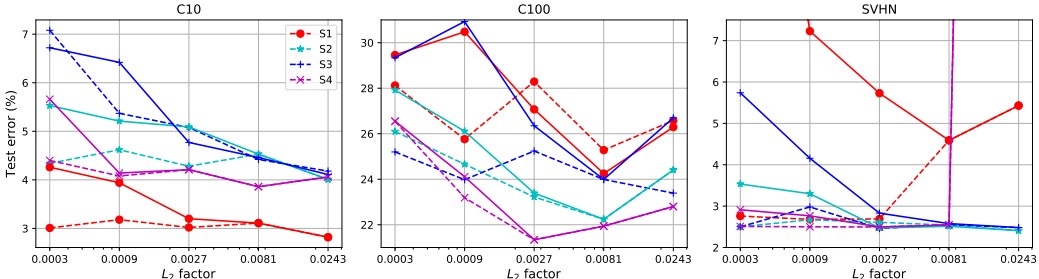

Figure 8: Effect of $L_2$ regularization of the inner objective during architecture search for DARTS (solid lines) and DARTS-ES (dashed).

Similar observations hold for disparity estimation on S6, where we vary the strength of standard data augmentation methods, such as shearing or brightness change, rather then masking parts of features, which is unreasonable for this task. The augmentation strength is described by an "augmentation scaling factor" (Appendix E). Table 2 summarizes the results. We report the average end point error (EPE), which is the Euclidean distance between the predicted and ground truth disparity maps. Data augmentation avoided the degenerate architectures and substantially improved results.

## 5.2 INCREASED $L_2$ REGULARIZATION

As a second type of regularization, we also tested different $L_2$ regularization factors $3i \cdot 10^{-4}$ for $i \in \{1, 3, 9, 27, 81\}$. Standard DARTS in fact does already include a small amount of $L_2$ regularization; $i = 1$ yields its default. Figure 8 shows that DARTS' test performance (solid lines) can be significantly improved by higher $L_2$ factors across all datasets and spaces, while keeping the dominant eigenvalue low (Figure 14). DARTS with early stopping (dashed lines) also benefits from additional regularization. Again, we observe the implicit regularization effect on the outer objective which reduces the overfitting of the architectural parameters. We again refer to Table 2 for disparity estimation; Appendix F shows similar results for language modelling (Penn TreeBank).

Table 2: Effect of regularization for disparity estimation. Search was conducted on FlyingThings3D (FT) and then evaluated on both FT and Sintel. Lower is better.

| Aug. Scale | Search model valid EPE | FT test EPE | Sintel test EPE | Params (M) |
|---|---|---|---|---|
| 0.0 | 4.49 | 3.83 | 5.69 | 9.65 |
| 0.1 | 3.53 | 3.75 | 5.97 | 9.65 |
| 0.5 | 3.28 | 3.37 | 5.22 | 9.43 |
| 1.0 | 4.61 | 3.12 | 5.47 | 12.46 |
| 1.5 | 5.23 | 2.60 | 4.15 | 12.57 |
| 2.0 | 7.45 | **2.33** | **3.76** | 12.25 |
| $L_2$ reg. factor | Search model valid EPE | FT test EPE | Sintel test EPE | Params (M) |
| $3 \times 10^{-4}$ | 3.95 | 3.25 | 6.13 | 11.00 |
| $9 \times 10^{-4}$ | 5.97 | **2.30** | 4.12 | 13.92 |
| $27 \times 10^{-4}$ | 4.25 | 2.72 | 4.83 | 10.29 |
| $81 \times 10^{-4}$ | 4.61 | **2.34** | **3.85** | 12.16 |

## 5.3 PRACTICAL ROBUSTIFICATION OF DARTS BY REGULARIZING THE INNER OBJECTIVE

Based on the insights from the aforementioned analysis and empirical results, we now propose two alternative simple modifications to make DARTS more robust in practice without having to manually tune its regularization hyperparameters.

**DARTS with adaptive regularization** One option is to adapt DARTS' regularization hyperparameters in an automated way, in order to keep the architectural weights in areas of the validation loss objective with smaller curvature. The simplest off-the-shelf procedure towards this desiderata would be to increase the regularization strength whenever the dominant eigenvalue starts increasing rapidly. Algorithm 1 (DARTS-ADA, Appendix D.1) shows such a procedure. We use the same stopping criterion as in DARTS-ES (Section 4.3), roll back DARTS to the epoch when this criterion is met, and continue the search with a larger regularization value $R$ for the remaining epochs (larger by a factor of $\eta$). This procedure is repeated whenever the criterion is met, unless the regularization value exceeds some maximum predefined value $R_{max}$.

**Multiple DARTS runs with different regularization strength** Liu et al. (2019) already suggested to run the search phase of DARTS four times, resulting in four architectures, and to return

the best of these four architectures w.r.t. validation performance when retrained from scratch for a limited number of epochs. We propose to use the same procedure, with the only difference that the four runs use different amounts of regularization. The resulting RobustDARTS (R-DARTS) method is conceptually very simple, trivial to implement and likely to work well if any of the tried regularization strengths works well.

Table 3 evaluates the performance of our practical robustifications of DARTS, DARTS-ADA and R-DARTS (based on either L2 or ScheduledDropPath regularization), by comparing them to the original DARTS, DARTS-ES and Random Search with weight sharing (RS-ws). For each of these methods, as proposed in the DARTS paper (Liu et al., 2019), we ran the search four independent times with different random seeds and selected the architecture used for the final evaluation based on a validation run as described above.

Table 3: Empirical evaluation of practical robustified versions of DARTS. Each entry is the test error after retraining the selected architecture as usual. The best method for each setting is boldface and underlined, the second best boldface.

| Benchmark | | RS-ws | DARTS | R-DARTS(DP) | R-DARTS(L2) | DARTS-ES | DARTS-ADA |
|-----------|-----|-------|-------|-------------|-------------|----------|-----------|
| C10 | S1 | 3.23 | 3.84 | 3.11 | **2.78** | **3.01** | 3.10 |
| | S2 | 3.66 | 4.85 | 3.48 | **3.31** | **3.26** | 3.35 |
| | S3 | 2.95 | 3.34 | 2.93 | **2.51** | 2.74 | **2.59** |
| | S4 | 8.07 | 7.20 | **3.58** | **3.56** | 3.71 | 4.84 |
| C100 | S1 | **23.30** | 29.46 | 25.93 | 24.25 | 28.37 | **24.03** |
| | S2 | **21.21** | 26.05 | 22.30 | **22.24** | 23.25 | 23.52 |
| | S3 | 23.75 | 28.90 | **22.36** | 23.99 | 23.73 | **23.37** |
| | S4 | 28.19 | 22.85 | 22.18 | **21.94** | **21.26** | 23.20 |
| SVHN | S1 | 2.59 | 4.58 | **2.55** | 4.79 | 2.72 | **2.53** |
| | S2 | 2.72 | 3.53 | **2.52** | **2.51** | 2.60 | 2.54 |
| | S3 | 2.87 | 3.41 | **2.49** | **2.48** | 2.50 | 2.50 |
| | S4 | 3.46 | 3.05 | 2.61 | **2.50** | 2.51 | **2.46** |

As the table shows, in accordance with Li & Talwalkar (2019), RS-ws often outperformed the original DARTS; however, with our robustifications, DARTS typically performs substantially better than RS-ws. DARTS-ADA consistently improved over standard DARTS for all benchmarks, indicating that a gradual increase of regularization during search prevents ending up in the bad regions of the architectural space. Finally, RobustDARTS yielded the best performance and since it is also easier to implement than DARTS-ES and DARTS-ADA, it is the method that we recommend to be used in practice.

Finally, since the evaluations in this paper have so far focussed on smaller subspaces of the original DARTS search space, the reader may wonder how well Robust-DARTS works on the full search spaces. As Table 4 shows, RobustDARTS performed similarly to DARTS for the two original benchmarks from the DARTS paper (PTB and CIFAR-10), on which DARTS was developed and is well tuned; however, even when only changing the dataset to CIFAR-100 or SVHN, RobustDARTS already performed significantly better than DARTS, underlining its robustness.

Table 4: DARTS vs. RobustDARTS on the original DARTS search spaces. We show mean ± stddev for 5 repetitions (based on 4 fresh subruns each as in Table 3); for the more expensive PTB we could only afford 1 such repetition.

| Benchmark | DARTS | R-DARTS(L2) |
|-----------|-------|-------------|
| C10 | **2.91 ± 0.25** | **2.95 ± 0.21** |
| C100 | 20.58 ± 0.44 | **18.01 ± 0.26** |
| SVHN | 2.46 ± 0.09 | **2.17 ± 0.09** |
| PTB | 58.64 | **57.59** |

## 6 Conclusions

We showed that the generalization performance of architectures found by DARTS is related to the eigenvalues of the Hessian matrix of the validation loss w.r.t. the architectural parameters. Standard DARTS often results in degenerate architectures with large eigenvalues and poor generalization. Based on this observation, we proposed a simple early stopping criterion for DARTS based on tracking the largest eigenvalue. Our empirical results also show that properly regularizing the inner objective helps controlling the eigenvalue and therefore improves generalization. Our findings substantially improve our understanding of DARTS' failure modes and lead to much more robust versions. They are consistent across many different search spaces on image recognition tasks and also for the very different domains of language modelling and disparity estimation. Our code is available for reproducibility.

### Acknowledgments

The authors acknowledge funding by the Robert Bosch GmbH, support by the European Research Council (ERC) under the European Unions Horizon 2020 research and innovation programme through grant no. 716721, and by BMBF grant DeToL.

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

# A   MORE DETAIL ON DARTS

Here we present a detailed description of DARTS architectural update steps. We firstly provide the general formalism which computes the gradient of the outer level problem in (1) by means of the implicit function theorem. Afterwards, we present how DARTS computes the gradient used to update the architectural parameters $\alpha$.

## A.1   DERIVATIVE WITH SMOOTHED NON-QUADRATIC LOWER LEVEL PROBLEM

Consider the general definition of the bi-level optimization problem as given by (1) and (2). Given that $f$ is twice continuously differentiable and that all stationary points are local minimas, one can make use of the implicit function theorem to find the derivative of the solution map $\theta^*(y)$ w.r.t. $y$ (Bengio, 2000). Under the smoothness assumption, the optimality condition of the lower level (2) is $\nabla_\theta f(y, \theta) = \mathbf{0}$, which defines an implicit function $\theta^*(y)$. With the assumption that $\min_\theta f(y, \theta)$ has a solution, there exists a $(y, \theta^*)$ such that $\nabla_\theta f(y, \theta^*) = \mathbf{0}$. Under the condition that $\nabla_\theta f(y, \theta^*) = \mathbf{0}$ is continuously differentiable and that $\theta^*(y)$ is continuously differentiable at $y$, implicitly differentiating the last equality from both sides w.r.t. y and applying the chain rule, yields:

$$\frac{\partial(\nabla_\theta f)}{\partial \theta}(y, \theta^*) \cdot \frac{\partial \theta^*}{\partial y}(y) + \frac{\partial(\nabla_\theta f)}{\partial y}(y, \theta^*) = \mathbf{0}. \tag{3}$$

Assuming that the Hessian $\nabla_\theta^2 f(y, \theta^*)$ is invertible, we can rewrite (3) as follows:

$$\frac{\partial \theta^*}{\partial y}(y) = -\left(\nabla_\theta^2 f(y, \theta^*)\right)^{-1} \cdot \frac{\partial(\nabla_\theta f)}{\partial y}(y, \theta^*). \tag{4}$$

Applying the chain rule to (1) for computing the total derivative of $F$ with respect to $y$ yields:

$$\frac{dF}{dy} = \frac{\partial F}{\partial \theta} \cdot \frac{\partial \theta^*}{\partial y} + \frac{\partial F}{\partial y}, \tag{5}$$

where we have omitted the evaluation at $(y, \theta^*)$. Substituting (4) into (5) and reordering yields:

$$\frac{dF}{dy} = \frac{\partial F}{\partial y} - \frac{\partial F}{\partial \theta} \cdot \left(\nabla_\theta^2 f\right)^{-1} \cdot \frac{\partial^2 f}{\partial \theta \partial y}. \tag{6}$$

equation 6 computes the gradient of $F$, given the function $\theta^*(y)$, which maps outer variables to the inner variables minimizing the inner problem. However, in most of the cases obtaining such a mapping is computationally expensive, therefore different heuristics have been proposed to approximate $dF/dy$ (Maclaurin et al., 2015; Pedregosa, 2016; Franceschi et al., 2017; 2018).

## A.2   DARTS ARCHITECTURAL GRADIENT COMPUTATION

DARTS optimization procedure is defined as a bi-level optimization problem where $\mathcal{L}_{valid}$ is the outer objective (1) and $\mathcal{L}_{train}$ is the inner objective (2):

$$\min_\alpha \mathcal{L}_{valid}(\alpha, w^*(\alpha)) \tag{7}$$

$$s.t. \quad w^*(\alpha) = \arg\min_w \mathcal{L}_{train}(\alpha, w), \tag{8}$$

where both losses are determined by both the architecture parameters $\alpha$ (outer variables) and the network weights $w$ (inner variables). Based on Appendix A.1, under some conditions, the total derivative of $\mathcal{L}_{valid}$ w.r.t. $\alpha$ evaluated on $(\alpha, w^*(\alpha))$ would be:

$$\frac{d\mathcal{L}_{valid}}{d\alpha} = \nabla_\alpha \mathcal{L}_{valid} - \nabla_w \mathcal{L}_{valid} \left(\nabla_w^2 \mathcal{L}_{train}\right)^{-1} \nabla_{\alpha,w}^2 \mathcal{L}_{train}, \tag{9}$$

where $\nabla_\alpha = \frac{\partial}{\partial \alpha}$, $\nabla_w = \frac{\partial}{\partial w}$ and $\nabla_{\alpha,w}^2 = \frac{\partial^2}{\partial \alpha \partial w}$. Computing the inverse of the Hessian is in general not possible considering the high dimensionality of the model parameters $w$, therefore resolving to gradient-based iterative algorithms for finding $w^*$ is necessary. However, this would also require to

optimize the model parameters $w$ till convergence each time $\alpha$ is updated. If our model is a deep neural network it is clear that this computation is expensive, therefore Liu et al. (2019) propose to approximate $w^*(\alpha)$ by updating the current model parameters $w$ using a single gradient descent step:

$$w^*(\alpha) \approx w - \xi \nabla_w \mathcal{L}_{train}(\alpha, w), \tag{10}$$

where $\xi$ is the learning rate for the virtual gradient step DARTS takes with respect to the model weights $w$. From equation 10 the gradient of $w^*(\alpha)$ with respect to $\alpha$ is

$$\frac{\partial w^*}{\partial \alpha}(\alpha) = -\xi \nabla^2_{\alpha, w} \mathcal{L}_{train}(\alpha, w), \tag{11}$$

By setting the evaluation point $w^* = w - \xi \nabla_w \mathcal{L}_{train}(\alpha, w)$ and following the same derivation as in Appendix A.1, we obtain the DARTS architectural gradient approximation:

$$\frac{d\mathcal{L}_{valid}}{d\alpha}(\alpha) = \nabla_\alpha \mathcal{L}_{valid}(\alpha, w^*) - \xi \nabla_w \mathcal{L}_{valid}(\alpha, w^*) \nabla^2_{\alpha, w} \mathcal{L}_{train}(\alpha, w^*), \tag{12}$$

where the inverse Hessian $\nabla^2_w \mathcal{L}^{-1}_{train}$ in (9) is replaced by the learning rate $\xi$. This expression however contains again an expensive vector-matrix product. Liu et al. (2019) reduce the complexity by using the finite difference approximation around $w^\pm = w \pm \epsilon \nabla_w \mathcal{L}_{valid}(\alpha, w^*)$ for some small $\epsilon = 0.01 / \|\nabla_w \mathcal{L}_{valid}(\alpha, w^*)\|_2$ to compute the gradient of $\nabla_\alpha \mathcal{L}_{train}(\alpha, w^*)$ with respect to $w$ as

$$\nabla^2_{\alpha, w} \mathcal{L}_{train}(\alpha, w^*) \approx \frac{\nabla_\alpha \mathcal{L}_{train}(\alpha, w^+) - \nabla_\alpha \mathcal{L}_{train}(\alpha, w^-)}{2\epsilon \nabla_w \mathcal{L}_{valid}(\alpha, w^*)} \qquad \Leftrightarrow$$

$$\nabla_w \mathcal{L}_{valid}(\alpha, w^*) \nabla^2_{\alpha, w} \mathcal{L}_{train}(\alpha, w^*) \approx \frac{\nabla_\alpha \mathcal{L}_{train}(\alpha, w^+) - \nabla_\alpha \mathcal{L}_{train}(\alpha, w^-)}{2\epsilon}. \tag{13}$$

In the end, combining equation 12 and equation 13 gives the gradient to compute the architectural updates in DARTS:

$$\frac{d\mathcal{L}_{valid}}{d\alpha}(\alpha) = \nabla_\alpha \mathcal{L}_{valid}(\alpha, w^*) - \frac{\xi}{2\epsilon} \big( \nabla_\alpha \mathcal{L}_{train}(\alpha, w^+) - \nabla_\alpha \mathcal{L}_{train}(\alpha, w^-) \big) \tag{14}$$

In all our experiments we always use $\xi = \eta$ (also called second order approximation in Liu et al. (2019)), where $\eta$ is the learning rate used in SGD for updating the parameters $w$.

## B    CONSTRUCTION OF S1 FROM SECTION 3

We ran DARTS two times on the default search space to find the two most important operations per mixed operation. Initially, every mixed operation consists of 8 operations. After the first DARTS run, we drop the 4 (out of 8) least important ones. In the second DARTS run, we drop the 2 (out of the remaining 4) least important ones. S1 is then defined to contain only the two remaining most important operations per mixed op. Refer to Figure 9 for an illustration of this pre-optimized space.

## C    FINAL ARCHITECTURE EVALUATION

Similar to the original DARTS paper (Liu et al., 2019), the architecture found during the search are scaled up by increasing the number of filters and cells and retrained from scratch to obtain the final test performance. For CIFAR-100 and SVHN we use 16 number of initial filters and 8 cells when training architectures from scratch for all the experiments we conduct. The rest of the settings is the same as in Liu et al. (2019).

On CIFAR-10, when scaling the ScheduledDropPath drop probability, we use the same settings for training from scratch the found architectures as in the original DARTS paper, i.e. 36 initial filters and 20 stacked cells. However, for search space S2 and S4 we reduce the number of initial filters to 16 in order to avoid memory issues, since the cells found with more regularization usually are composed only with separable convolutions. When scaling the $L_2$ factor on CIFAR-10 experiments we use 16 initial filters and 8 stacked cells, except the experiments on S1, where the settings are the same as in Liu et al. (2019), i.e. 36 initial filters and 20 stacked cells.

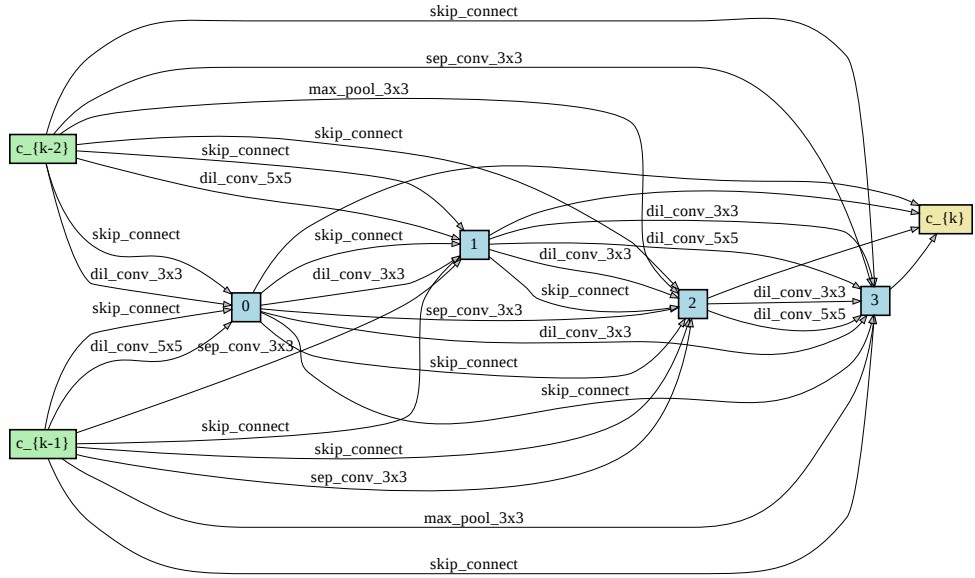

(a) Normal cell space

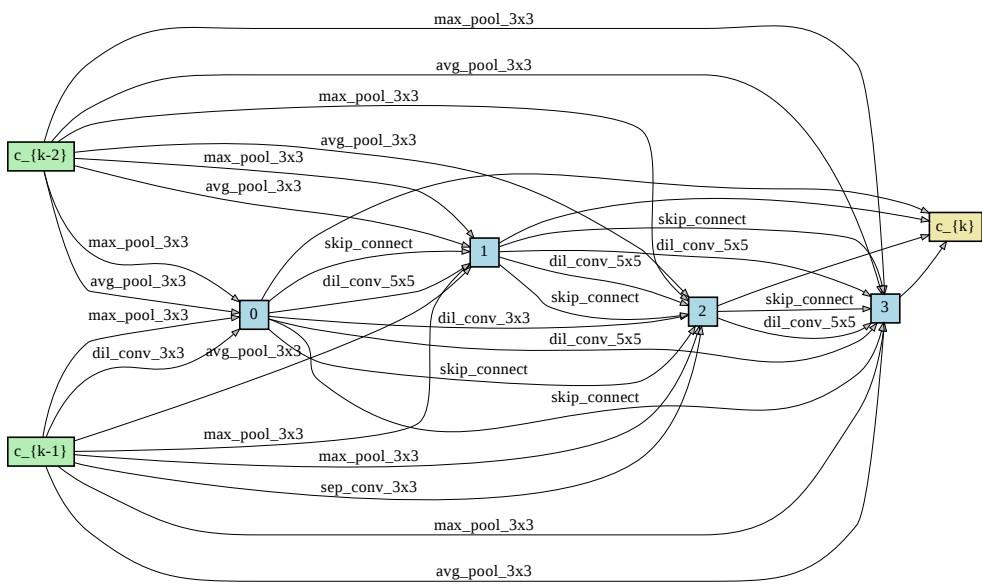

(b) Reduction cell space

Figure 9: Search space **S1**.

Note that although altering the regularization factors during DARTS search, when training the final architectures from scratch we always use the same values for them as in Liu et al. (2019), i.e. ScheduledDropPath maximum drop probability linearly increases from 0 towards 0.2 throughout training, Cutout is always enabled with cutout probability 1.0, and the $L_2$ regularization factor is set to $3 \cdot 10^{-4}$.

# D    ADDITIONAL EMPIRICAL RESULTS

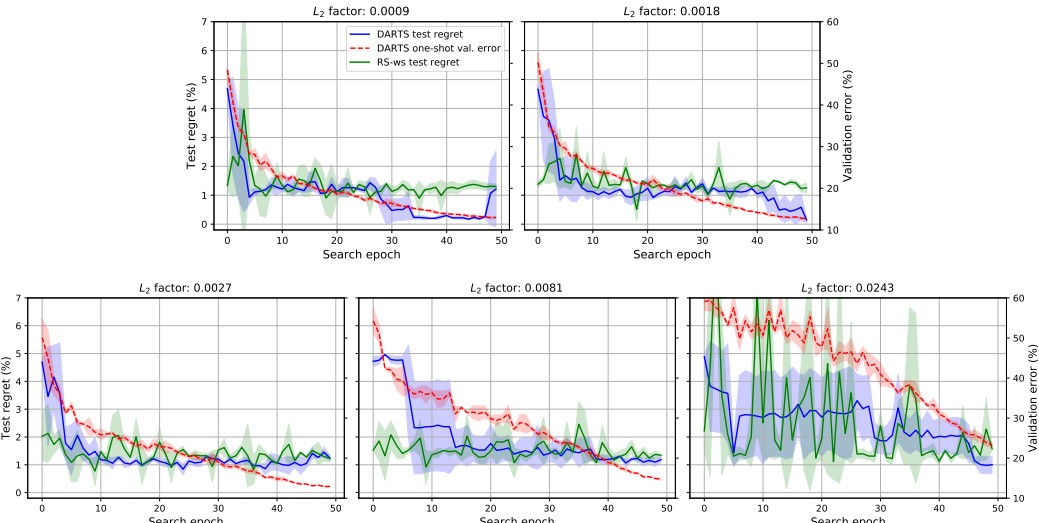

Figure 10: Test regret and validation error of the search (one-shot) model when running DARTS on S5 and CIFAR-10 with different $L_2$ regularization values. The architectural parameters' overfit reduces as we increase the $L_2$ factor and successfully finds the global minimum. However, we notice that the architectural parameters start underfitting as we increase to much the $L_2$ factor, i.e. both validation and test error increase.

Table 5: Validation (train) and test accuracy on CIFAR-10 of the search and final evaluation models, respectively. The values in the last column show the maximum eigenvalue $\lambda_{max}^{\alpha}$ (computed on a random sampled mini-batch) of the Hessian, at the end of search for different maximum drop path probability). The four blocks in the table state results for the search spaces S1-S4, respectively.

| | Drop Prob. | Valid acc. | | | Test acc. | | | Params | | | $\lambda_{max}^{\alpha}$ | | |
|---|---|---|---|---|---|---|---|---|---|---|---|---|---|
| | | seed 1 | seed 2 | seed 3 | seed 1 | seed 2 | seed 3 | seed 1 | seed 2 | seed 3 | seed 1 | seed 2 | seed 3 |
| S1 | 0.0 | 87.22 | 87.01 | 86.98 | 96.16 | 94.43 | 95.43 | 2.24 | 1.93 | 2.03 | 1.023 | 0.835 | 0.698 |
| | 0.2 | 84.24 | 84.32 | 84.22 | 96.39 | 96.66 | 96.20 | 2.63 | 2.84 | 2.48 | 0.148 | 0.264 | 0.228 |
| | 0.4 | 82.28 | 82.18 | 82.79 | 96.44 | **96.94** | 96.76 | 2.63 | 2.99 | 3.17 | 0.192 | 0.199 | 0.149 |
| | 0.6 | 79.17 | 79.18 | 78.84 | **96.89** | 96.93 | **96.96** | 3.38 | 3.02 | 3.17 | 0.300 | 0.255 | 0.256 |
| S2 | 0.0 | 88.49 | 88.40 | 88.35 | 95.15 | 95.48 | 96.11 | 0.93 | 0.86 | 0.97 | 0.684 | 0.409 | 0.268 |
| | 0.2 | 85.29 | 84.81 | 85.36 | 95.15 | 95.40 | 96.14 | 1.28 | 1.44 | 1.36 | 0.270 | 0.217 | 0.145 |
| | 0.4 | 82.03 | 82.66 | 83.20 | 96.34 | **96.50** | 96.44 | 1.28 | 1.28 | 1.36 | 0.304 | 0.411 | 0.282 |
| | 0.6 | 79.86 | 80.19 | 79.70 | **96.52** | 96.35 | 96.29 | 1.21 | 1.28 | 1.36 | 0.292 | 0.295 | 0.281 |
| S3 | 0.0 | 88.78 | 89.15 | 88.67 | 94.70 | 96.27 | 96.66 | 2.21 | 2.43 | 2.85 | 0.496 | 0.535 | 0.446 |
| | 0.2 | 85.61 | 85.60 | 85.50 | 96.78 | 96.84 | **96.74** | 3.62 | 4.04 | 2.99 | 0.179 | 0.185 | 0.202 |
| | 0.4 | 83.03 | 83.24 | 83.43 | **97.07** | **96.85** | 96.48 | 4.10 | 3.74 | 3.38 | 0.156 | 0.370 | 0.184 |
| | 0.6 | 79.86 | 80.03 | 79.68 | 96.91 | 94.56 | 96.44 | 4.46 | 2.30 | 2.66 | 0.239 | 0.275 | 0.280 |
| S4 | 0.0 | 86.33 | 86.72 | 86.46 | 92.80 | 93.22 | 93.14 | 1.05 | 1.13 | 1.05 | 0.400 | 0.442 | 0.314 |
| | 0.2 | 81.01 | 82.43 | 82.03 | 95.84 | 96.08 | 96.15 | 1.44 | 1.44 | 1.44 | 0.070 | 0.054 | 0.079 |
| | 0.4 | 79.49 | 79.67 | 78.96 | 96.11 | 96.30 | 96.28 | 1.44 | 1.44 | 1.44 | 0.064 | 0.057 | 0.049 |
| | 0.6 | 74.54 | 74.74 | 74.37 | **96.42** | **96.36** | **96.64** | 1.44 | 1.44 | 1.44 | 0.057 | 0.060 | 0.066 |

## D.1    ADAPTIVE DARTS DETAILS

We evaluated DARTS-ADA (Section 5.3) with $R = 3 \cdot 10^{-4}$ (DARTS default), $R_{max} = 3 \cdot 10^{-2}$ and $\eta = 10$ on all the search spaces and datasets we use for image classification. The results are shown in Table 3 (DARTS-ADA). The function `train_and_eval` conducts the normal DARTS search for one epoch and returns the architecture at the end of that epoch's updates and the `stop` value if a decision was made to stop the search and rollback to `stop_epoch`.

---

**Algorithm 1:** `DARTS_ADA`

---

```
/* E: epochs to search; R:  initial regularization value; R_max:  maximal
   regularization value; stop_criter:  stopping criterion; η:
   regularization increase factor                                        */
```
**Input** : E, $R$, $R_{max}$, `stop_criter`, $\eta$

```
/* start search for E epochs                                             */
```
**for** *epoch* **in** $E$ **do**
  ```/* run DARTS for one epoch and return stop=True together with the```
    `stop_epoch                                                        */`
  ```/* and the architecture at stop_epoch if the criterion is met       */```
  `stop, stop_epoch, arch ← train_and_eval(stop_criter);`
  **if** $stop$ & $R \leq R_{max}$ **then**
    ```/* start DARTS from stop_epoch with a larger R                  */```
    `arch ← DARTS_ADA(E - stop_epoch, η · R, R_max, stop_criter, η);`
    `break`
  **end**
**end**

**Output:** `arch`

---

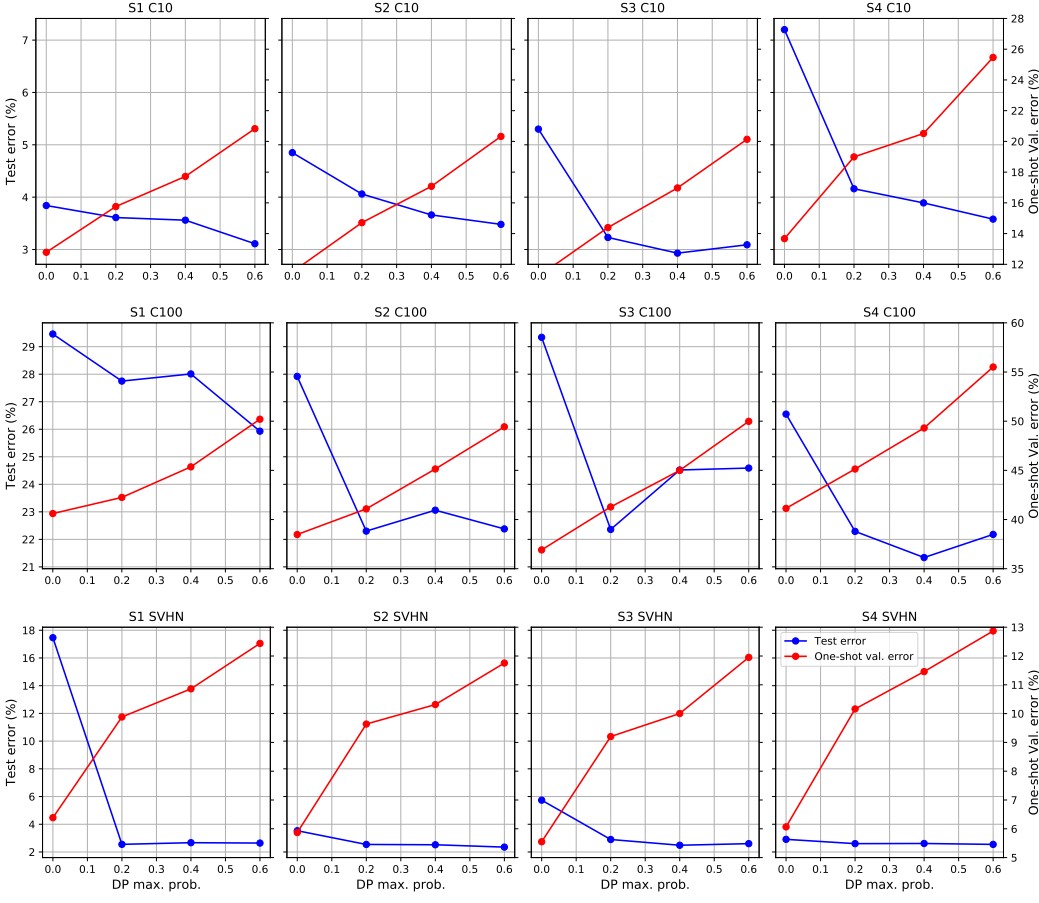

Figure 11: Test errors of architectures along with the validation error of the search (one-shot) model for each dataset and space when scaling the ScheduledDropPath drop probability. Note that these results (blue lines) are the same as the ones in Figure 8.

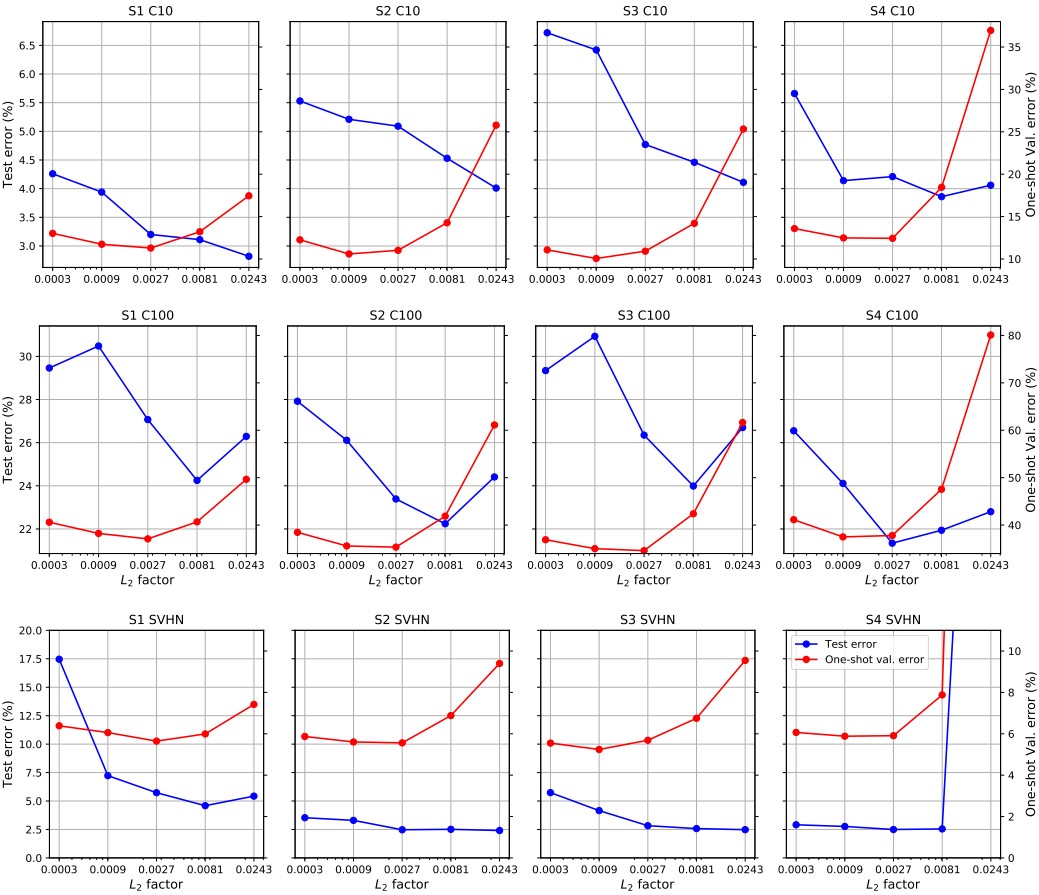

Figure 12: Test errors of architectures along with the validation error of the search (one-shot) model for each dataset and space when scaling the $L_2$ factor. Note that these results (blue lines) are the same as the ones in Figure 7.

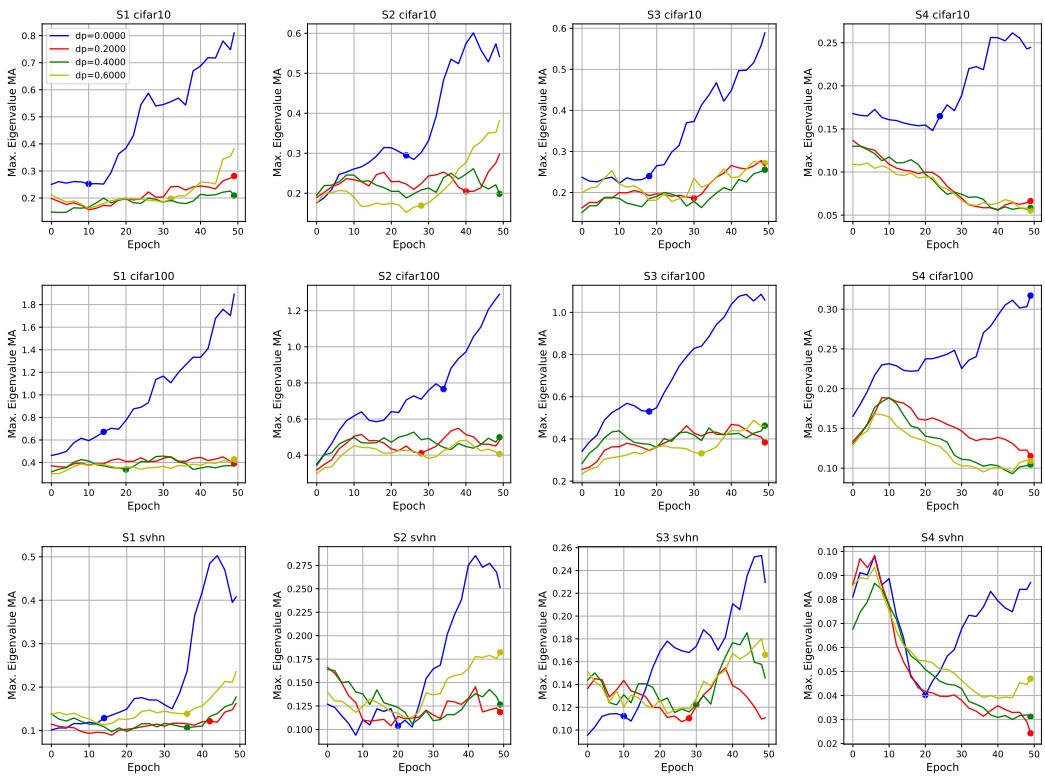

Figure 13: Local average of the dominant EV $\lambda_{max}^{\alpha}$ throughout DARTS search (for different drop path prob. values). Markers denote the early stopping point based on the criterion in Section 4.3.

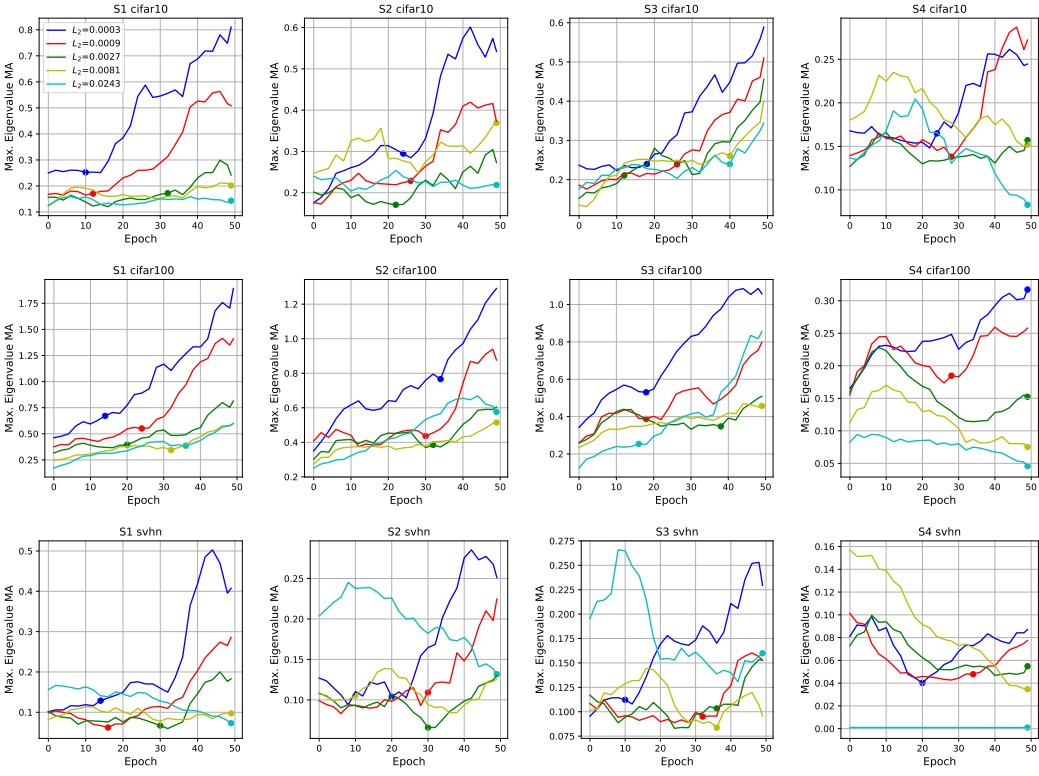

Figure 14: Effect of $L_2$ regularization no the EV trajectory. The figure is analogous to Figure 13.

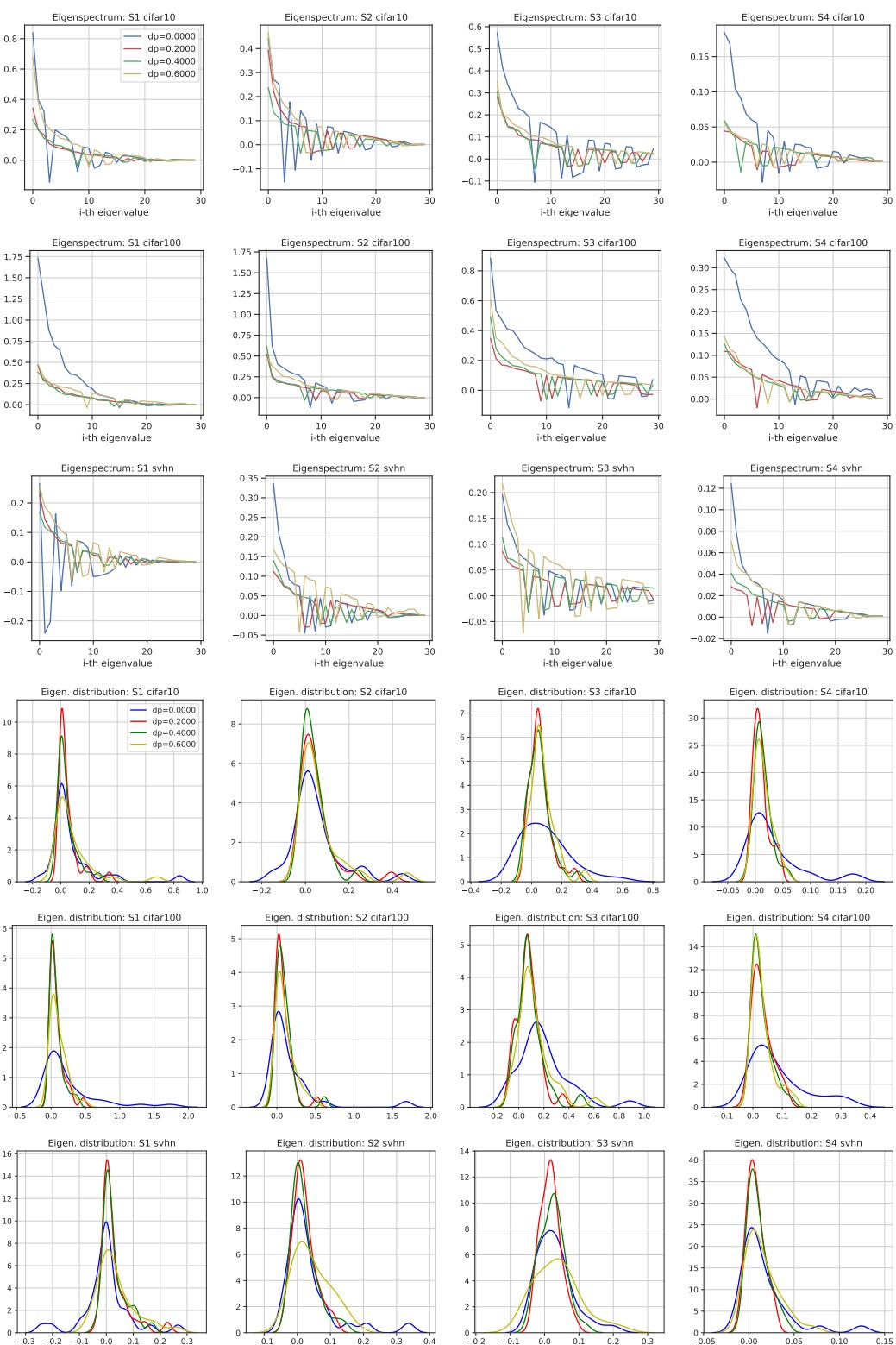

Figure 15: Effect of ScheduledDropPath and Cutout on the full eigenspectrum of the Hessian at the end of architecture search for each of the search spaces. Since most of the eigenvalues after the 30-th largest one are almost zero, we plot only the largest (based on magnitude) 30 eigenvalues here. We also provide the eigenvalue distribution for these 30 eigenvalues. Notice that not only the dominant eigenvalue is larger when $dp = 0$ but in general also the others.

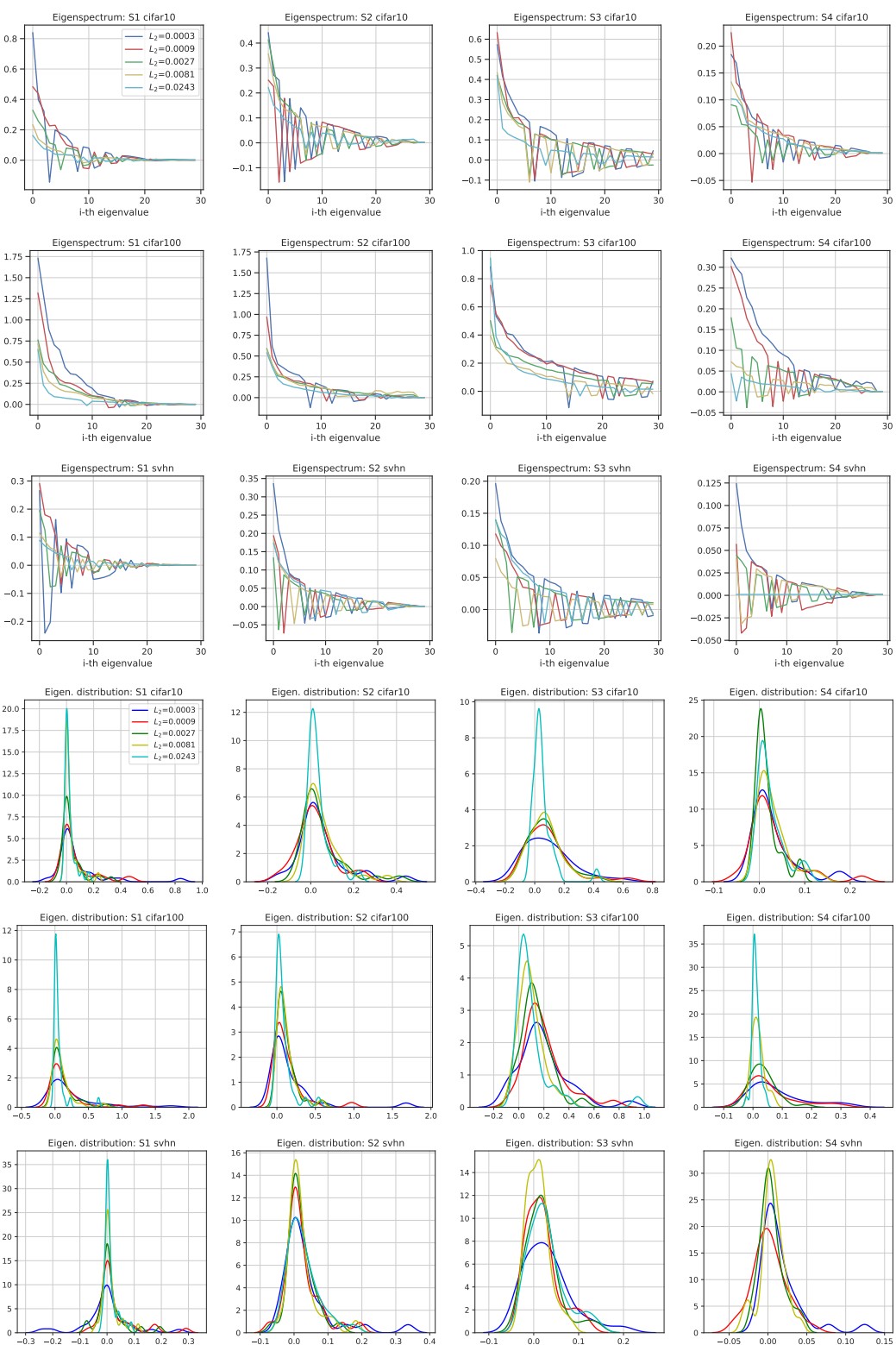

Figure 16: Effect of $L_2$ regularization on the full eigenspectrum of the Hessian at the end of architecture search for each of the search spaces. Since most of the eigenvalues after the 30-th largest one are almost zero, we plot only the largest (based on magnitude) 30 eigenvalues here. We also provide the eigenvalue distribution for these 30 eigenvalues. Notice that not only the dominant eigenvalue is larger when $L_2 = 3 \cdot 10^{-4}$ but in general also the others.

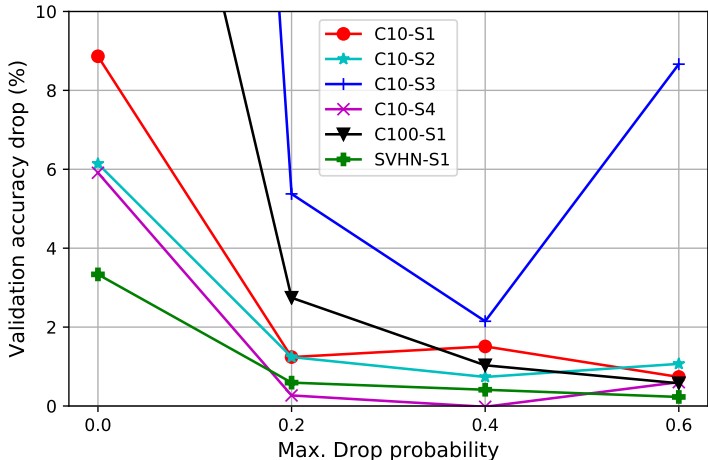

Figure 17: Drop in accuracy after discretizing the search model for different spaces, datasets and drop path regularization strengths.. Example of some of the settings from Section 5.

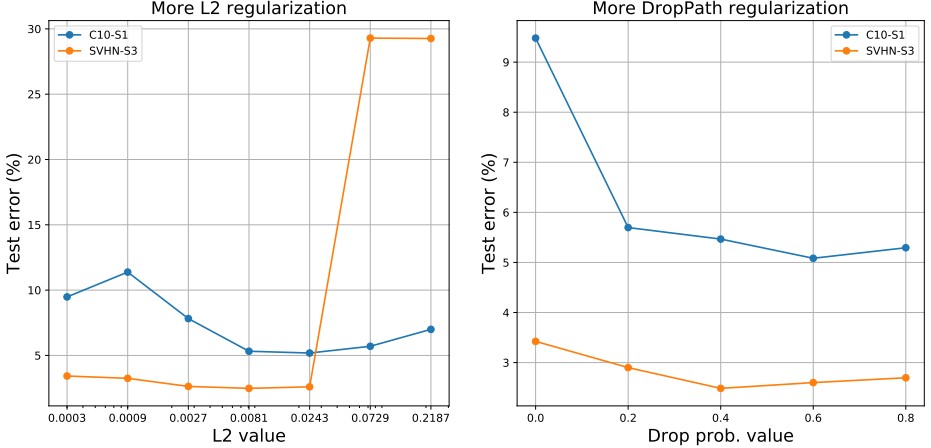

Figure 18: Effect of more regularization on the performance of found architectures by DARTS.

Table 6: Performance of architectures found by DARTS (-ES / -ADA) vs. RandomNAS with weight sharing. For each of the settings we repeat the search 3 times and report the mean ± std of the 3 found architectures retrained from scratch.

| Setting | | RandomNAS | DARTS | DARTS-ES | DARTS-ADA |
|---------|-----|-----------|-------|----------|-----------|
| C10 | S1 | $3.17 \pm 0.15$ | $4.66 \pm 0.71$ | $3.05 \pm 0.07$ | $\mathbf{3.03} \pm 0.08$ |
| | S2 | $3.46 \pm 0.15$ | $4.42 \pm 0.40$ | $\mathbf{3.41} \pm 0.14$ | $3.59 \pm 0.31$ |
| | S3 | $\mathbf{2.92} \pm 0.04$ | $4.12 \pm 0.85$ | $3.71 \pm 1.14$ | $2.99 \pm 0.34$ |
| | S4 | $89.39 \pm 0.84$ | $6.95 \pm 0.18$ | $4.17 \pm 0.21$ | $\mathbf{3.89} \pm 0.67$ |
| C100 | S1 | $25.81 \pm 0.39$ | $29.93 \pm 0.41$ | $28.90 \pm 0.81$ | $\mathbf{24.94} \pm 0.81$ |
| | S2 | $\mathbf{22.88} \pm 0.16$ | $28.75 \pm 0.92$ | $24.68 \pm 1.43$ | $26.88 \pm 1.11$ |
| | S3 | $24.58 \pm 0.61$ | $29.01 \pm 0.24$ | $26.99 \pm 1.79$ | $\mathbf{24.55} \pm 0.63$ |
| | S4 | $30.01 \pm 1.52$ | $24.77 \pm 1.51$ | $23.90 \pm 2.01$ | $\mathbf{23.66} \pm 0.90$ |
| SVHN | S1 | $2.64 \pm 0.09$ | $9.88 \pm 5.50$ | $2.80 \pm 0.09$ | $\mathbf{2.59} \pm 0.07$ |
| | S2 | $\mathbf{2.57} \pm 0.04$ | $3.69 \pm 0.12$ | $2.68 \pm 0.18$ | $2.79 \pm 0.22$ |
| | S3 | $2.89 \pm 0.09$ | $4.00 \pm 1.01$ | $2.78 \pm 0.29$ | $\mathbf{2.58} \pm 0.07$ |
| | S4 | $3.42 \pm 0.04$ | $2.90 \pm 0.02$ | $2.55 \pm 0.15$ | $\mathbf{2.52} \pm 0.06$ |

## D.2 A CLOSER LOOK AT THE EIGENVALUES

Over the course of all experiments from the paper, we tracked the largest eigenvalue across all configuration and datasets to see how they evolve during the search. Figures 13 and 14 shows the results across all the settings for image classification. It can be clearly seen that increasing the inner objective regularization, both in terms of $L_2$ or data augmentation, helps controlling the largest eigenvalue and keeping it to a small value, which again helps explaining why the architectures found with stronger regularization generalize better. The markers on each line highlight the epochs where DARTS is early stopped. As one can see from Figure 4, there is indeed some correlation between the average dominant eigenvalue throughout the search and the test performance of the found architectures by DARTS.

Figures 15 and 16 (top 3 rows) show the full spectrum (sorted based on eigenvalue absolute values) at the end of search, whilst bottom 3 rows plot the distribution of eigenvalues in the eigenspectrum. As one can see, not only the dominant eigenvalue is larger compared to the cases when the regularization is stronger and the generalization of architectures is better, but also the other eigenvalues in the spectrum have larger absolute value, indicating a sharper objective landscape towards many dimensions. Furthermore, from the distribution plots note the presence of more negative eigenvalues whenever the architectures are degenerate (lower regularization value) indicating that DARTS gets stuck in a point with larger positive and negative curvature of the validation loss objective, associated with a more degenerate Hessian matrix.

# E DISPARITY ESTIMATION

## E.1 DATASETS

We use the FlyingThings3D dataset (Mayer et al., 2016) for training AutoDispNet. It consists of rendered stereo image pairs and their ground truth disparity maps. The dataset provides a training and testing split consisting of $21,818$ and $4248$ samples respectively with an image resolution of $960 \times 540$. We use the Sintel dataset ( Butler et al. (2012)) for testing our networks. Sintel is another synthetic dataset from derived from an animated movie which also provides ground truth disparity maps (1064 samples) with a resolution of $1024 \times 436$.

## E.2 TRAINING

We use the AutoDispNet-C architecture as described in Saikia et al. (2019). However, we use the smaller search which consists of three operations: $MaxPool3 \times 3$, $SepConv3 \times 3$, and $SkipConnect$. For training the search network, images are downsampled by a factor of two and trained for $300k$ mini-batch iterations. During search, we use SGD and ADAM to optimize the inner and outer objectives respectively. Differently from the original AutoDispNet we do not warm-start the search model weights before starting the architectural parameter updates. The extracted network is also trained for $300k$ mini-batch iterations but full resolution images are used. Here, ADAM is used for optimization and the learning rate is annealed to $0$ from $1e-4$, using a cosine decay schedule.

## E.3 EFFECT OF REGULARIZATION ON THE INNER OBJECTIVE

To study the effect of regularization on the inner objective for AutoDispNet-C we use experiment with two types of regularization: data augmentation and of $L2$ regularization on network weights.

We note that we could not test the early stopping method on AutoDispNet since AutoDispNet relies on custom operations to compute feature map correlation (Dosovitskiy et al., 2015) and resampling, for which second order derivatives are currently not available (which are required to compute the Hessian).

**Data augmentation.** Inspite of fairly large number of training samples in FlyingThings3D, data augmentation is crucial for good generalization performance. Disparity estimation networks employ spatial transformations such as translation, cropping, shearing and scaling. Additionally, appearance transformations such as additive Gaussian noise, changes in brightness, contrast, gamma and color

are also applied. Parameters for such transformations are sampled from a uniform or Gaussian distribution (parameterized by a mean and variance). In our experiments, we vary the data augmentation strength by multiplying the variance of these parameter distributions by a fixed factor, which we dub the *augmentation scaling factor*. The extracted networks are evaluated with the same augmentation parameters. The results of increasing the augmentation strength of the inner objective can be seen in Table 2. We observe that as augmentation strength increases DARTS finds networks with more number of parameters and better test performance. The best test performance is obtained for the network with maximum augmentation for the inner objective. At the same time the search model validation error increases when scaling up the augmentation factor, which again enforces the argument that the overfitting of architectural parameters is reduced by this implicit regularizer.

**L2 regularization.** We study the effect of increasing regularization strength on the weights of the network. The results are shown in Table 2. Also in this case best test performance is obtained with the maximum regularization strength.

## F    RESULTS ON PENN TREEBANK

Here we investigate the effect of more $L_2$ regularization on the inner objective for searching recurrent cells on Penn Treebank (PTB). We again used a reduced search space with only *ReLU* and *identity* mapping as possible operations. The rest of the settings is the same as in (Liu et al., 2019).

We run DARTS search four independent times with different random seeds, each with four $L_2$ regularization factors, namely $5 \times 10^{-7}$ (DARTS default), $15 \times 10^{-7}$, $45 \times 10^{-7}$ and $135 \times 10^{-7}$. Figure 19 shows the test perplexity of the architectures found by DARTS with the aforementioned $L_2$ regularization values. As we can see, a stronger regularization factor on the inner objective makes the search procedure more robust. The median perplexity of the discovered architectures gets better as we increase the $L_2$ factor from $5 \times 10^{-7}$ to $45 \times 10^{-7}$, while the search model (one-shot) validation mean perplexity increases. This observation is similar to the ones on image classification shown in Figure 10, showing again that properly regularizing the inner objective helps reduce overfitting the architectural parameters.

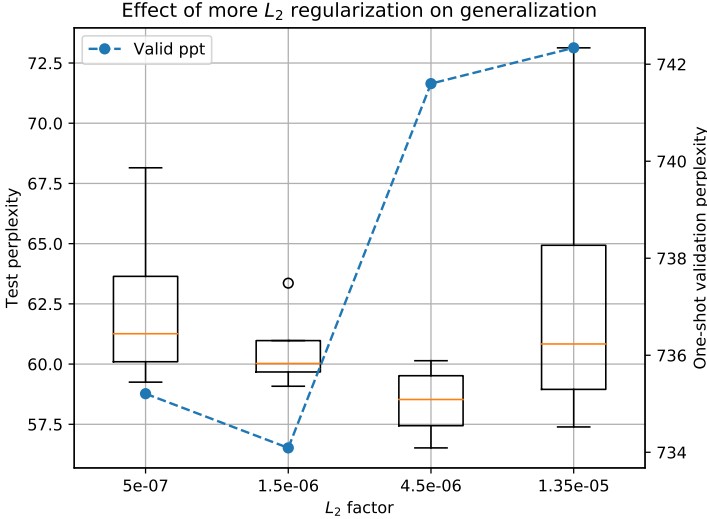

Figure 19: Performance of recurrent cells found with different $L_2$ regularization factors on the inner objective on PTB. We run DARTS 4 independent times with different random seeds, train each of them from scratch with the evaluation settings for 1600 epochs and report the median test perplexity. The blue dashed line denotes the validation perplexity of the search model.

# G  DISCOVERED CELLS ON SEARCH SPACES S1-S4 FROM SECTION 3 ON OTHER DATASETS

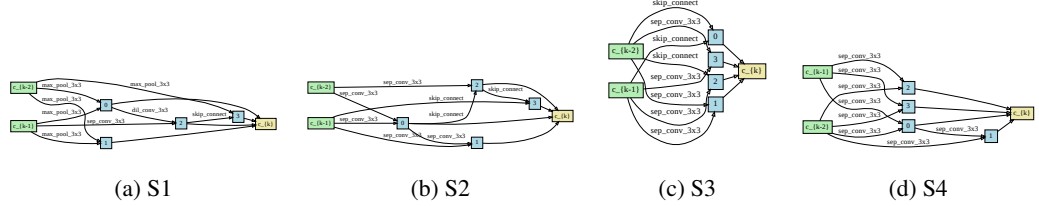

(a) S1        (b) S2        (c) S3        (d) S4

Figure 20: Reduction cells found by DARTS when ran on CIFAR-10 with its default hyperparameters on spaces S1-S4. These cells correspond with the normal ones in Figure 1.

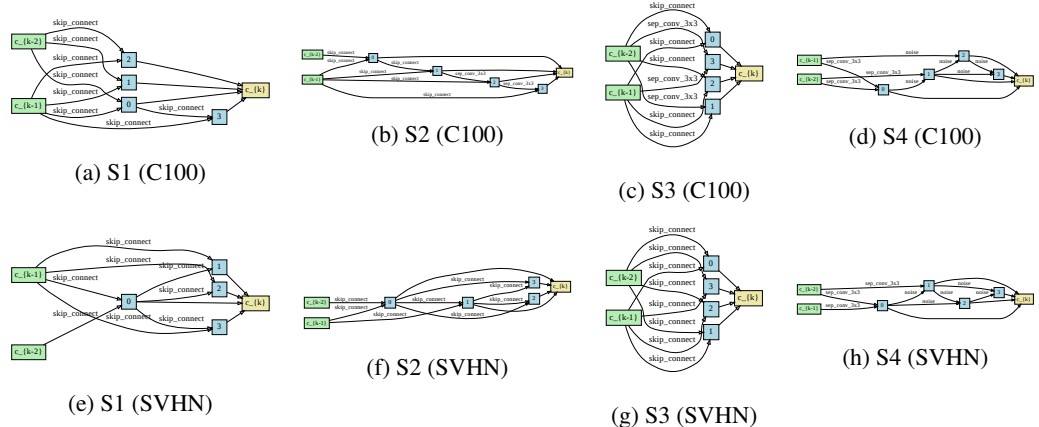

(a) S1 (C100)    (b) S2 (C100)    (c) S3 (C100)    (d) S4 (C100)

(e) S1 (SVHN)    (f) S2 (SVHN)    (g) S3 (SVHN)    (h) S4 (SVHN)

Figure 21: Normal cells found by DARTS on CIFAR-100 and SVHN when ran with its default hyperparameters on spaces S1-S4. Notice the dominance of parameter-less operations such as *skip connection* and *pooling* ops.

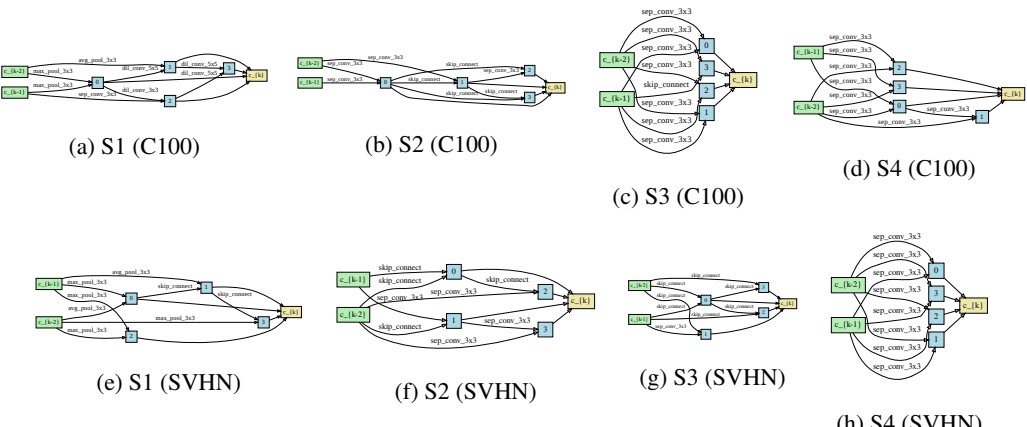

(a) S1 (C100)    (b) S2 (C100)    (c) S3 (C100)    (d) S4 (C100)

(e) S1 (SVHN)    (f) S2 (SVHN)    (g) S3 (SVHN)    (h) S4 (SVHN)

Figure 22: Reduction cells found by DARTS on CIFAR-100 and SVHN when ran with its default hyperparameters on spaces S1-S4.

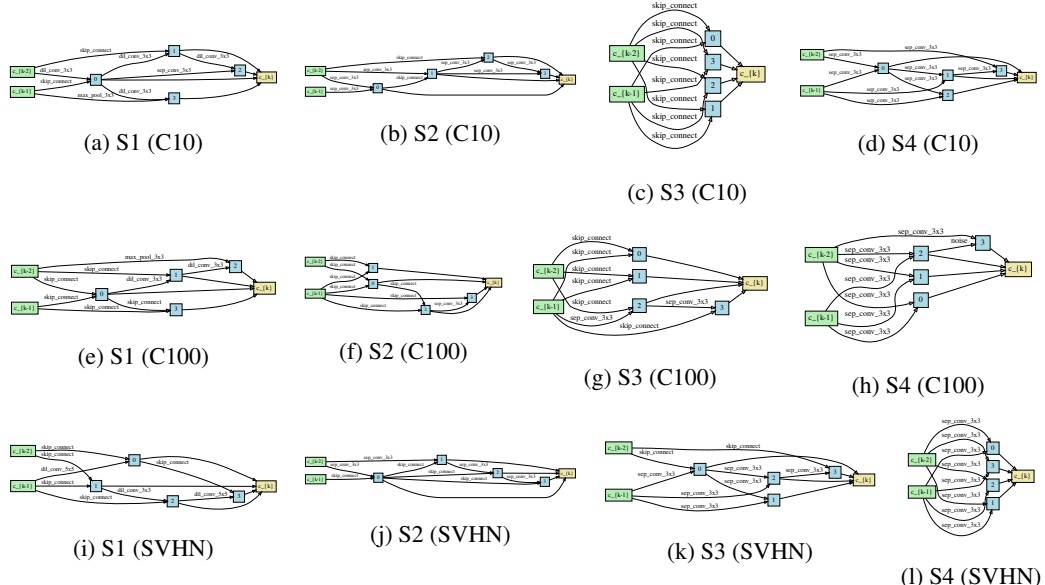

Figure 23: Normal cells found by DARTS-ES when ran with DARTS default hyperparameters on spaces S1-S4.

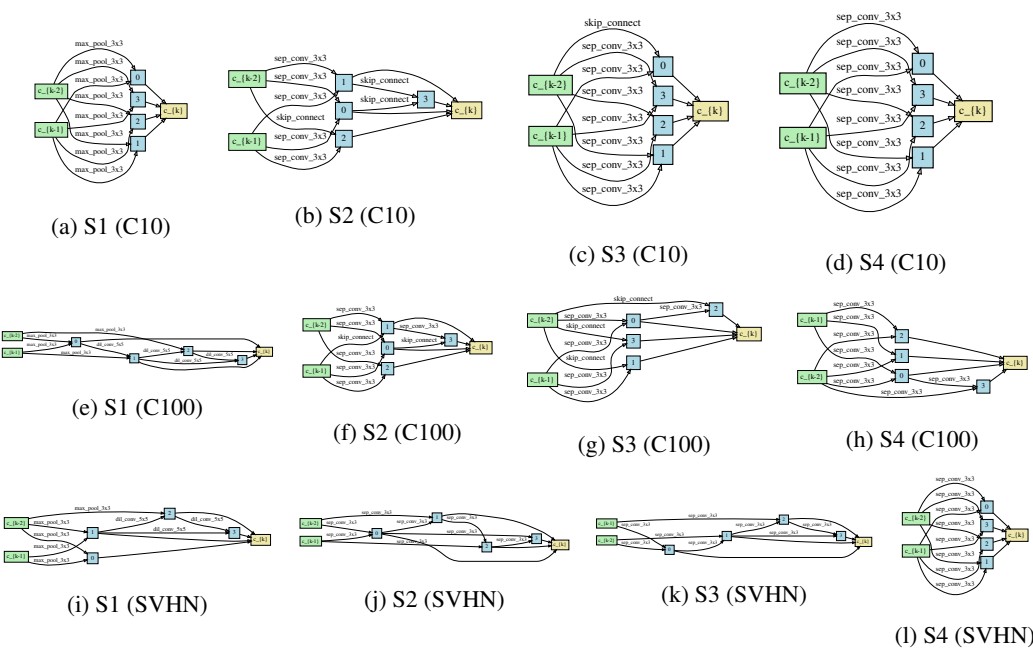

Figure 24: Reduction cells found by DARTS-ES when ran with DARTS default hyperparameters on spaces S1-S4.

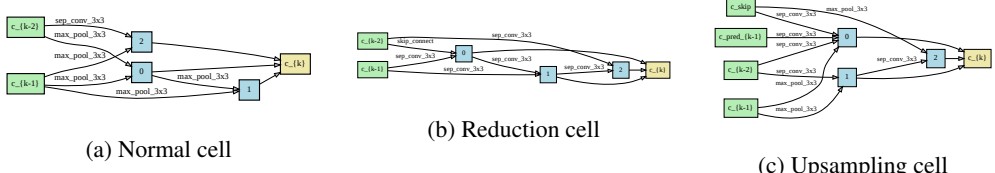

(a) Normal cell

(b) Reduction cell

(c) Upsampling cell

Figure 25: Cells found by AutoDispNet when ran on S6-d. These cells correspond to the results for augmentation scale 0.0 of Table 2.

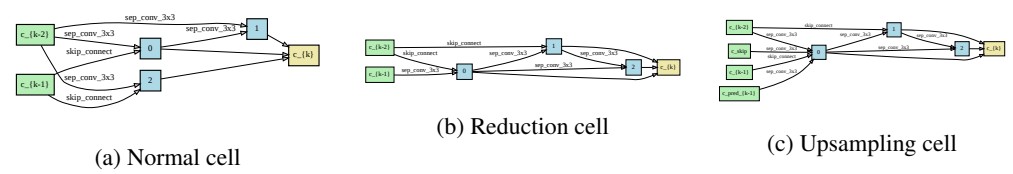

(a) Normal cell

(b) Reduction cell

(c) Upsampling cell

Figure 26: Cells found by AutoDispNet when ran on S6-d. These cells correspond to the results for augmentation scale 2.0 of Table 2.

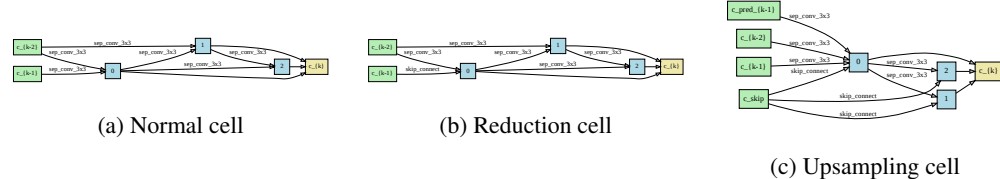

(a) Normal cell

(b) Reduction cell

(c) Upsampling cell

Figure 27: Cells found by AutoDispNet when ran on S6-d. These cells correspond to the results for $L_2 = 3 \cdot 10^{-4}$ of Table 2.

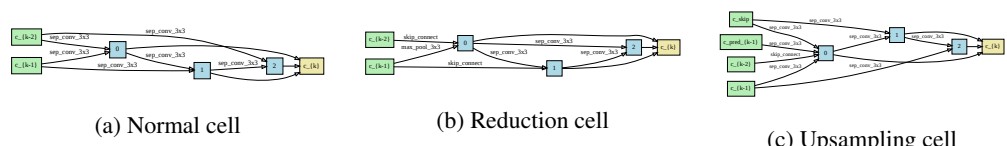

(a) Normal cell

(b) Reduction cell

(c) Upsampling cell

Figure 28: Cells found by AutoDispNet when ran on S6-d. These cells correspond to the results for $L_2 = 81 \cdot 10^{-4}$ of Table 2.

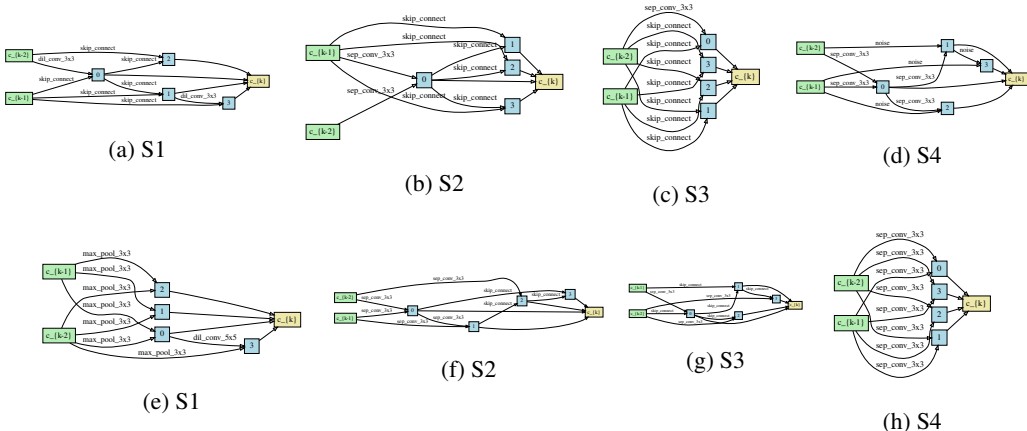

Figure 29: Normal (top row) and reduction (bottom) cells found by DARTS on CIFAR-10 when ran with its default hyperparameters on spaces S1-S4. Same as Figure 1 but with different random seed (seed 2).

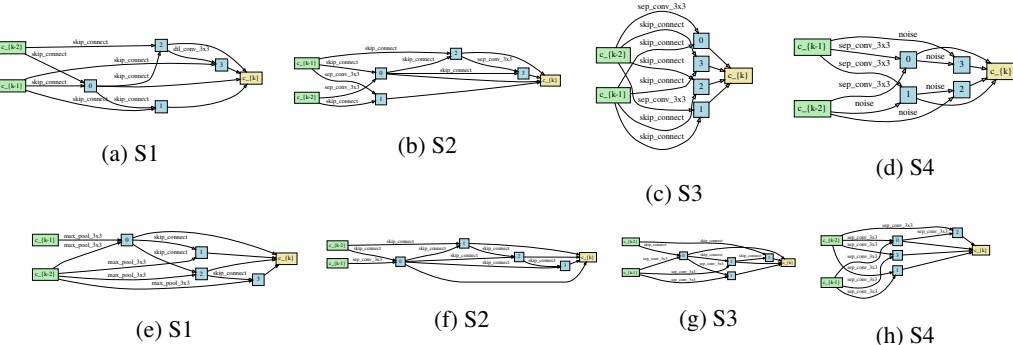

Figure 30: Normal (top row) and reduction (bottom) cells found by DARTS on CIFAR-10 when ran with its default hyperparameters on spaces S1-S4. Same as Figure 1 but with different random seed (seed 3).

