# OpenReview forum: "Understanding and Robustifying Differentiable Architecture Search"
_ICLR.cc/2020/Conference — Accept (Talk)_

### Official Review · AnonReviewer3 · 2019-10-23
**Official Blind Review #3**

**Rating:** 8

**Review:**

----- Updated after rebuttal period ---

The author's detailed response effectively addressed my concerns. I am moving my score to Accept. This paper proposes an interesting systematic study of differentiable approach in NAS.

------ Original Review ----

Summary

This paper presents a systematic evaluation on top of differentiable architecture search (DARTS) algorithm and shows it usually searched an architecture with all skip-connection. It empirically reveals that the largest eigenvalue of the Hessian matrix (\lambda) of loss w.r.t. architecture parameters has a strong correlation with the generalization ability (via loss of test dataset), and shows this \lambda will first decrease but then drastically increase after a certain epoch number on 4 different search spaces. It then proposes an early-stop scheme (DARTS-ES) to stop the search before this phenomenon occurs. In addition, it proposes to use data-augmentation, path-dropping and tuning L2 regularization during the search, namely Robust-DARTS(R-DARTS), and yield constantly better results over original DARTS on 3 datasets.


Overall, the observation that the largest eigenvalue of the Hessian matrix is novel and intriguing, and the experiments are extensive and meaningful. The idea to use more search spaces for comparison is fair and performance increase demonstrates the proposed R-DARTS and DARTS-ES  are effective. Although I still have some questions regarding the detail settings, I think this paper provides a novel angle to understand the search phase of DARTS, and proposed simple but effective regularization can be beneficial to the research community using DARTS.


Main concerns:

- Problem of DARTS as a motivation
The claims of local smoothness/sharpness and generalization are related to network generalization is quite intriguing, however, only using largest eigenvalue of Hessian matrix as an indicator of this local shape does not seem to be enough. A recent paper on loss-landscape visualization [1] provides means to examine this hypothesis directly, and could the author try to provide additional visualization to support their claim? Otherwise, the paper's claim does not generalize to the local shape of the loss function, and should stays with the largest eigenvalue. It is totally okay in my perspective, but just indicates some revision to the main text and analysis of Section 4.2.

- Questions about Figure 3 experiments
How is test error computed? Is it on a batch of test-split, or the entire one? Also, which architecture is used to compute this test error? Paper mentioned, in Section 4.1, the word "final architecture", but does this refer to the super-net (the one-shot model in paper's definition), or the stand-alone model obtained via binarized architecture alphas? If latter, is this generalization error obtained from training from scratch? Or simply using the super-net parameters during the search? Since the conclusion of this plot serves as the foundation of designing R-DARTS and DARTS-ES, if the experiments are only conducted over a small set of images or the binarized model with super-net parameters, it undercuts the credibility of the conclusion, largest architectural eigenvalue, and generalization ability.


- More independent runs of experiments.
In Figure 3, validation of original DARTS, and Table 1, DARTS vs DARTS-ES, paper runs the experiment for 3 times and take the average, but for the proposed R-DARTS, it is not. Is there a reason why not scaling the experiments? I suggest the author provide results over 5 runs, like the one in Table 4 for PTB and show if the R-DARTS truly surpasses DARTS constantly. This question also applies to Figure 1, when paper claims the original DARTS found poor cell type, could this be repetitive over multiple runs?

I guess for the experiments in Table 3, it is already done since paper mentioned the reported results are the best model of searching 4 times.

- R-DARTS failed to out-perform DARTS in the original space on CIFAR-10
This is confusing, will this suggest, if tuning well, DARTS will surpass R-DARTS(L2) in other cases as well? Since this is the only setting that DARTS is built upon.


Minor comments and questions

- Using test data during search
After showing the strong correlation between the largest eigenvalue of Hessian and the network generalization error, the early-stop is natural, however, does this mean the model selection is using the test data? Or the actual test-data is never seen during the search phase of Section 4.3.

- L2 stabilizes max eigenvalue
Paper uses L2 coefficient up to 0.0243, showing constant improvement of test error while validation error drops in CIFAR-10 of Figure 11. Could the author try larger coefficients to determine when this trend will stop?

- Question about section 4.2
Performance drop due to the binarized operation (pruning step) in DARTS analysis is very interesting, I am curious how many architectures does the paper evaluate in Figure 5, when the dominant eigenvalue is smaller than 0.25? Since the conclusion is "low curvature never led to large performance drops" if the number of points is too few, it is not that convincing, especially from the plot, we see at eigenvalue = 0.5, there exists 2 architecture with >20% drop. In addition, what does each point in Figure 5 refer to? The best model (and the binarized one according to the argmax of \alpha) of one independent DARTS run or some binarized models sampled from a distribution on top of the same DARTS run (meaning only one super-net)? Is this experiment follows the setting in Figure 4?

- Figure 6, C10 S2, DARTS-ES is worse than DARTS when Drop probability = 0.6, whereas all other cases, DARTS-ES outperforms DARTS, why does this happen? Could the author comment on it?

- ScheduledDropPath in section 5.1
Does Drop-path belongs to data-augmentation techniques? It is more like a regularization in my perspective and should be grouped with 5.2.

- S1 S2... in Table 3
Does this refer to the search space? Or different random seed (mentioned )

- one-shot v.s. weight sharing model
One-shot in NAS domain is firstly introduced by Bender et al., while Pham et al. use parameter sharing. The reason to use one-shot is that all the sub-paths will have a fair chance to be trained.

- Typos
1. In section 2.1, line 4 "better.Similarly" should have space.

--- Reference ---
[1] Li et al., Visualizing the Loss Landscape of Neural Nets, arxiv'17.

**Experience Assessment:**

I have published one or two papers in this area.

**Review Assessment: Checking Correctness Of Derivations And Theory:**

I carefully checked the derivations and theory.

**Review Assessment: Checking Correctness Of Experiments:**

I carefully checked the experiments.

**Review Assessment: Thoroughness In Paper Reading:**

I read the paper thoroughly.

---

> ### Author Response · Authors · 2019-11-15
> **Author response 1/3 to official blind review #3**
>
> Many thanks for your very detailed and very useful review, your positive feedback, and for the acceptance score. We have updated the paper and now reply to all your questions in detail.
>
> Q1: Problem of DARTS as a motivation [Using the largest eigenvalue does not seem enough as an indicator of the local shape?]
> A1: We agree with the reviewer, the largest eigenvalue by itself is not enough. We did indeed also compute the full Hessian eigenspectrum on a randomly sampled validation mini-batch (please see Figure 14 and Figure 15 in Appendix D.2), and as one can see, not only the dominant eigenvalue is larger when comparing a low regularization factor vs. a high regularization factor, but also the other eigenvalues throughout the spectrum. This indicates that the curvature is higher not only towards one principal axis, but towards all the principal axes. The distribution of the eigenvalues in the eigenspectrum show clearly that for lower regularization factors the tail of the distribution becomes larger. We also thank the reviewer for pointing us to the very interesting paper on loss-landscape visualization; we are currently working on integrating this into our code.
>
>
> Q2: Questions about Figure 3 experiments
> A2: All test errors reported in the paper are indeed computed on the full test set, using the final stand-alone model (the single architecture we find in the end), obtained via applying the argmax to the optimized architectural weights; for computing test errors, we always train these models from scratch. The word “final architecture” in Section 4.1 refers to this final stand-alone model -- thanks, we updated the paper to make this clearer. The super-net parameters are only used for the results in Figure 5 in order to compute the correlation between the accuracy drop after binarization (we call this discretization in our paper) and the dominant eigenvalues of the Hessian.
>
>
> Q3:  More independent runs of experiments.
> A3: We actually do 4 search runs for each R-DARTS run in Table 4. We use the following procedure (Protocol 1) introduced in the DARTS paper [1] to select the architecture that will be trained from scratch using the evaluation settings:
> Do 4 independent DARTS (R-DARTS) search runs with the same (4 different) regularization factor(s).
> Retrain from scratch the 4 found architectures for 100 epochs and select the best according to the validation accuracy.
> Retrain from scratch the selected architecture (with more initial filters, stacked cells, etc.) for 600 epochs and compute the test error on the full test data.
> For the image classification datasets in Table 4 we repeated this protocol 5 times and report the mean +/- std test error of the 5 architectures returned from step 3. For the results in Table 3 and PTB in Table 4 we only repeated this protocol once due to the large computational costs of doing this for a lot of cases, however each of these entries is already based on 4 independent DARTS runs, and the best selected model is always retrained from scratch.
>
> The results in Figure 3 and Table 1 report the results when using the following simpler procedure (Protocol 2):
> Do 3 independent DARTS search runs with the same regularization factor.
> Retrain from scratch the 3 found architectures using the full evaluation pipeline (more initial filters, more stacked cells, etc.) and compute the test error on the full test data.
> Report the mean +/- std test error of the 3 architectures.
>
> For completeness, we added Table 6 (we had these results already before the rebuttal phase) in the Appendix, which reports the results when running Protocol 2 using Random Search with Weight Sharing [2], DARTS [1], DARTS-ES and DARTS-ADA for all the settings in Table 3. Note that the 3 architectures evaluated in Table 6 are a subset of the 4 architectures used in Protocol 1. We also added in Appendix G (Figures 27 and 28) the cells found when running the experiments in Figure 1 with 2 other random seeds; the qualitative results indeed remain the same.

---

> > ### Author Response · Authors · 2019-11-15
> > **Author response 2/3 to official blind review #3**
> >
> > Q4: “R-DARTS failed to out-perform DARTS in the original space on CIFAR-10
> > This is confusing, will this suggest, if tuning well, DARTS will surpass R-DARTS(L2) in other cases as well? Since this is the only setting that DARTS is built upon.”
> >
> > A4: We agree with the reviewer that it indeed seems that the default DARTS hyperparameters are well-tuned for CIFAR-10, the dataset used during the development of DARTS. As Table 3 indicates, only changing the dataset leads to sub-optimal behaviour of DARTS. Of course one could tune DARTS’ hyperparameters on each new benchmark, but much care would have to be taken to avoid this getting too expensive in a practical setting. Furthermore, tuning weight-sharing NAS algorithms is not that straightforward, because (as we show) the search model validation performance does not correlate with the generalization of the architectures that DARTS finds. Therefore, one needs to optimize the error of stand-alone architectures (retrained from scratch) evaluated on a separate validation set (different from the subset used for updating the architectural parameters during the search), which is one of the most computationally expensive parts. R-DARTS solves this problem by only altering one hyperparameter internally and still has the same computational costs as running 1 DARTS run with the protocol 1 above as suggested in the DARTS paper..
> >
> >
> > Q5: Using test data during search?
> > A5: We *never* use the test data when conducting any of the proposed heuristics. The test data is only used to indicate the generalization of the found architectures retrained from scratch in the very end, *after the search has finished*. It is in fact precisely the advantage of our early stopping mechanism based on curvature information that we do *not* need a separate dataset for the early stopping.
> >
> >
> > Q6: L2 stabilizes max eigenvalue [Could the author try larger coefficients to determine when this trend will stop?]
> > A6: Thank you for raising this question. We would expect the generalization error of the found architectures to drop when increasing above a certain value. This is confirmed by the plots we added to Appendix H. For those we used two of our benchmarks, S1-C10 and S3-SVHN and conducted the DARTS search 3 independent times for each L2 value and for each DropPath max. probability value. The results show the mean test error when retraining from scratch using the same settings as in Fig. 6 and Fig. 7 of these 3 architectures.
> >
> >
> > Q7:  Question about section 4.2
> > A7: The x axis in Figure 5 shows the dominant eigenvalue after the search has finished, while the y axis shows the absolute value of the difference: “search model validation accuracy” - “binarized model (according to the argmax of \alpha) validation accuracy, evaluated using the search model weights, not retrained from scratch”. Note that the settings for each point in the plot are different: the plot includes all our runs across all our search spaces (S1-S4) and datasets (C10, C100, SVHN). The points in Figure 4 on the other hand show the dominant eigenvalue on the x axis and the test error of the found architectures by DARTS when retrained from scratch. In this case we only use the C10 S1 settings (in total 24 points in the plot: 3 seeds x (4 runs with dp \in {0, 0.2, 0.4, 0.6} + 4 runs with L_2 \in {0.0009, 0.0027, 0.0081, 0.0243}).
> >
> > Note that what we consider a “low” curvature depends on the dataset and search space, since, as, e.g., Figure 12 shows, the scale of the dominant eigenvalue differs across these: e.g., for S2-SVHN the blue line goes up to a value of 0.275, while for S2-C100 it goes up to 1.3. This means that a ‘high’ or ‘low’ eigenvalue actually is relative to the benchmark we evaluate our algorithm.

---

> > > ### Author Response · Authors · 2019-11-15
> > > **Author response 3/3 to official blind review #3**
> > >
> > > Q8: Figure 6, C10 S2
> > > A8: We believe this behaviour comes from the inherent noise present in the eigenvalues of the Hessian when computing it on a validation mini-batch. Due to this noise, the stopping criterion would work less well in rare cases. E.g., when the architectural parameters do not overfit on the validation data, but the noise in the Hessian nevertheless triggers the early stopping mechanism too early, and therefore the resulting architecture is worse than the one DARTS would have found if it did not early stop. This case is more likely when running DARTS with a higher regularization value than its default values. In general, as shown in Figure 6 and Figure 7, the gain when early stopping with regularization values different from the default, is smaller (or negative as in the example you mentioned: C10 S2 with drop prob=0.6) compared the gain when early stopping DARTS with its default regularization factor.
> > >
> > >
> > > Q9: ScheduledDropPath in section 5.1
> > > A9: Thanks, we agree. We nevertheless listed it in the augmentation section because whenever we enable ScheduledDropPath during search we also increase the Cutout probability linearly from 0 to 1 throughout the search alongside the ScheduledDropPath. We now clarified this in the paper.
> > >
> > >
> > > Q10:  S1 S2... in Table 3
> > > A10: S{1, 2, 3, 4} in Table 3 indeed refers to the different search spaces as described in Section 3.
> > >
> > > Q11: one-shot v.s. weight sharing model
> > > A11: We agree with the reviewer that these two nomenclatures may have conceptually different meanings. So far, we have been using “one-shot” and “weight-sharing” interchangeably, but taking into account that “one-shot” can be seen as subset of “weight-sharing” methods, we changed this in our paper and now refer to the search model as weight-sharing model. But we are open to suggestions concerning nomenclature.
> > >
> > > - Typos
> > > Thank you for reading our paper in detail. We fixed this typo.
> > >
> > > -- References --
> > > [1] Hanxiao Liu, Karen Simonyan, Yiming Yang, DARTS: Differentiable Architecture Search, ICLR 2019
> > > [2] Liam Li, Ameet Talwalkar. Random Search and Reproducibility for Neural Architecture Search, UAI 2019

---

### Official Review · AnonReviewer1 · 2019-10-24
**Official Blind Review #1**

**Rating:** 8

**Review:**

This paper seeks to understand why Differential Architecture Search (DAS) might fail to find neural net architectures that perform well. The authors perform a series of experiments using different kinds of search spaces and datasets, and concluded that a major culprit is the discretization/pruning step at the end of DARTS. To avoid this, the authors propose early stopping based on measuring the eigenvalue of the Hessian of the validation loss. The results look promising (though as someone who is not familiar with the datasets, I don't have a sense of the significance of improvements.)

In general, this is a strong paper. I enjoyed reading it. It describes the problem clearly and performs a set of convincing experiments to support the claims. I especially like how different constrained search spaces are investigated, as this makes the results easier to interpret. I think the analysis in this paper will benefit researchers who work on similar problems.



**Experience Assessment:**

I do not know much about this area.

**Review Assessment: Checking Correctness Of Derivations And Theory:**

N/A

**Review Assessment: Checking Correctness Of Experiments:**

I assessed the sensibility of the experiments.

**Review Assessment: Thoroughness In Paper Reading:**

N/A

---

> ### Author Response · Authors · 2019-11-15
> **Author response to official blind review #1**
>
> Many thanks for your very positive feedback and acceptance score!

---

### Official Review · AnonReviewer4 · 2019-10-30
**Official Blind Review #4**

**Rating:** 8

**Review:**

This paper studies the causes of why DARTS often results in models that do not generalize to the test set. The paper finds that DARTS models do not generalize due to sharp minima, partially caused by the discretizing step in DARTS. The paper presents many experiments and studies on multiple search spaces, showing that this problem is general. To address this problem, the paper proposes several different ways to address this, e.g., an early stopping criteria and regularization methods.

Overall, the paper is well written, thorough experiments (various tasks and search spaces) show the benefit of the approach.  The final experiments using the full DARTS space also show an improvement over standard DARTS.

The final method is fairly simple, running the search with different regularization parameters and keeping the best model, which suggests it could be widely used for DARTS-based approaches.

 Two (very) minor comments:
- There's a missing space before "Similarly" in section 2.1
- Extra ")" in last paragraph of section 2.2 (after the argmin eq).

**Experience Assessment:**

I have published one or two papers in this area.

**Review Assessment: Checking Correctness Of Derivations And Theory:**

N/A

**Review Assessment: Checking Correctness Of Experiments:**

I carefully checked the experiments.

**Review Assessment: Thoroughness In Paper Reading:**

I read the paper thoroughly.

---

> ### Author Response · Authors · 2019-11-15
> **Author response to official blind review #4**
>
> Many thanks for your very positive feedback and for the acceptance score! We fixed the two typos you pointed out.

---

### Public Comment · ~James_Smith10 · 2019-09-29
**Another work which early stops DARTS to stabilize the performance**

I remember that there is another paper doing the similar thing recently (DARTS+: Improved Differentiable Architecture Search with Early Stopping, https://arxiv.org/abs/1909.06035).

They also point out the instability of DARTS and try to use the early stopping trick to improve it. However, they early stop DARTS when the ranking of conv operations becomes stable which is quite different from yours. I wonder if there is any underlying connection between the two works, and which one is more fundamental.

They claim that they achieve the state-of-the-art results of classification on several datasets, including CIFAR-10, CIFAR-100 and ImageNet. Do you have evaluated your algorithms on ImageNet? Thanks!

---

> ### Author Response · Authors · 2019-10-01
> **Comments on other work with early stopping**
>
> Hi James,
> thank you for your comment. Indeed, the independent work you mention also proposes an early stopping for DARTS to avoid instabilities. Their observation seems - to some extent - to be consistent with ours. E.g., they also observe an increased prevalence of skip connections over the course of search, resulting in poorly performing architectures and motivating an early stopping criterion.
>
> However, there are some fundamental differences in both works:
> 1) The DARTS+ authors hypothesize that the problem in DARTS is solving the bi-level optimization problem. In contrast, we show that the bi-level optimization problem is actually solved fine and that the problem instead lies in the discretization and generalization. E.g., in Figure 3 in our paper, you can see (left plot), that the validation error (= objective to be minimized in the bi-level optimization problem) steadily decreases, i.e., the alternating optimization of DARTS works fine in these cases. Rather, only the test error (middle plot) increases. Therefore, we argue that the problem lies in the generalization capability. Unfortunately, it is not clear if the DARTS+ authors plot the train-, validation-, or test error in their Figure 2 (showing the collapse), but based on our finding we’d guess it is test error, which would then be consistent with our experiments.
> 2) The DARTS+ authors hypothesize that changing hyperparameters (such as regularization parameters) will likely not solve the problem. However, in our experiments, with the fixed search time of 50 epochs (which is the vanilla DARTS default), we observed that increasing regularization strength *did* help to prevent poor generalization performance (on average). We refer to Figures 6 and 7 in our paper, which show that increasing drop path probabilities and L2 regularization, respectively, usually helps to find better architectures -- also without early stopping, i.e., with vanilla DARTS (solid lines in all plots). See also Table 2 as well as Figures 10 and 11 in the appendix.
> 3) Early stopping is only a minor contribution in our work that resulted as a byproduct from one of our core scientific contributions: analysing the eigenvalues of the Hessian of the validation loss w.r.t. the architectural parameters and relating them to generalization performance.
>
> Having said this, the proposed stopping criterion in DARTS+ based on the ranking of architectural parameters seems interesting; we will look into this in the future and investigate whether, e.g., the ranking also correlates with large eigenvalues.
>
> We did not run our models on ImageNet yet, as the focus of our work is on gaining insights why DARTS fails and how we can prevent these failures. Therefore, we rather went the opposite direction: to *small* cases that ought to be easy for NAS methods and which are computationally relatively cheap, so that we can do multiple repeats and really gain fundamental scientific insights about the performance of different NAS methods, rather than engineering gains on ImageNet. In that vein, we introduce (and make available code for) 12 new, relatively cheap, benchmarks where DARTS fails that ought to help provide a solid foundation for empirical work in studying this problem henceforth.
> In contrast, the DARTS+ paper is definitely very strong on the engineering side, achieving new state-of-the-art performance for CIFAR by using a final training pipeline that is modified from the original DARTS paper (e.g., they train for 2000 epochs rather than 600 on CIFAR, use AutoAugment, and emphasize their use of various further tricks). We commend the DARTS+ authors for their great work on improving the final training pipeline -- it is amazing to see how much they can push the SOTA by doing so. On the other hand, these changes are orthogonal to the architecture search and would likely similarly improve the performance of vanilla DARTS. This makes it hard to assess how much the DARTS+ search actually improves over the DARTS search and how much improvement is due to the different final training pipeline. We refer to the recent NAS best practice paper ( https://arxiv.org/abs/1909.02453 ), which argues that for a fair comparison between NAS methods, the final training pipeline (optimizer, hyperparameters, data augmentation, number of epochs, ...) should be identical. For this reason, to ensure a fair comparison, we did not change this final training pipeline at all in our work (but only changed the architecture search process). If the DARTS+ authors release the code for their final training pipeline for CIFAR and ImageNet we could also run DARTS and our extensions of it on that.
>
> Thank you for your interest, and we’ll be happy to answer further questions.
> The authors

---

### Decision · Program_Chairs · 2019-12-19

**Decision:**

Accept (Talk)

**Comment:**

This paper studies the properties of Differentiable Architecture Search, and in particular when it fails, and then proposes modifications that improve its performance for several tasks. The reviews were all very supportive with three Accept opinions, and authors have addressed their comments and suggestions. Given the unanimous reviews, this appears to be a clear Accept.